# Novel C1q receptor-mediated signaling controls neural stem cell behavior and neurorepair

Francisca Benavente[1,2,3]*, Katja M Piltti[1,4], Mitra J Hooshmand[1,4], Aileen A Nava[1], Anita Lakatos[1,4], Brianna G Feld[1,5], Dana Creasman[1,2], Paul D Gershon[6,7], Aileen Anderson[1,2,4]*

[1]Sue and Bill Gross Stem Cell Research Center, Irvine, United States; [2]Department of Anatomy and Neurobiology, Irvine, United States; [3]Center of Regenerative Medicine, Facultad de Medicina, Universidad del Desarrollo, Santiago, Chile; [4]Institute for Memory Impairments and Neurological Disorders, Irvine, United States; [5]Bridges to Stem Cell Research Program (BSCR), California State University, Long Beach, United States; [6]Department of Physical Medicine and Rehabilitation, Irvine, United States; [7]Department of Molecular Biology & Biochemistry, UC-Irvine, Irvine, United States

*For correspondence:
fbenaven@uci.edu (FB);
aja@uci.edu (AA)

Competing interests: The authors declare that no competing interests exist.

**Abstract** C1q plays a key role as a recognition molecule in the immune system, driving autocatalytic complement cascade activation and acting as an opsonin. We have previously reported a non-immune role of complement C1q modulating the migration and fate of human neural stem cells (hNSC); however, the mechanism underlying these effects has not yet been identified. Here, we show for the first time that C1q acts as a functional hNSC ligand, inducing intracellular signaling to control cell behavior. Using an unbiased screening strategy, we identified five transmembrane C1q signaling/receptor candidates in hNSC (CD44, GPR62, BAI1, c-MET, and ADCY5). We further investigated the interaction between C1q and CD44 , demonstrating that CD44 mediates C1q induced hNSC signaling and chemotaxis in vitro, and hNSC migration and functional repair in vivo after spinal cord injury. These results reveal a receptor-mediated mechanism for C1q modulation of NSC behavior and show that modification of C1q receptor expression can expand the therapeutic window for hNSC transplantation.

## Introduction

Therapeutic transplantation of human neural stem cells (hNSC) offers a promising approach for neural repair in neurodegenerative disorders and central nervous system (CNS) injuries. While the immunomodulatory effect of donor stem cells on the host has been extensively studied (*Tena and Sachs, 2014*; *Pluchino et al., 2005*; *Zhang et al., 2013*) the converse effect of the host immune-microenvironment on donor stem cells has received little attention. We have previously shown that polymorphonuclear neutrophils (PMNs), which infiltrate the spinal cord at acute time points post trauma (*Beck et al., 2010*), alter the responses of donor cells after acute spinal cord injury (SCI) transplantation. Specifically, systemic immunodepletion of PMNs inhibits donor hNSC astrogliogenesis and rescues the capacity of donor cells to promote functional repair after acute transplantation into the SCI microenvironment (*Nguyen et al., 2017*). These data demonstrate that functional integration of transplanted stem cells is dependent, at least in part, on interactions between donor cells and cellular/molecular cues in the host microenvironment. We also demonstrated that secreted factors derived from two distinct immune populations, PMN and macrophages/microglia (Mφ), drive hNSC migration and lineage selection (fate) and identified complement C1q and C3a as molecular

mediators (*Hooshmand et al., 2017*). These data highlight the importance of cues from the host inflammatory microenvironment in modulating NSC behavior, and identify a significant in vitro and in vivo role for complement C1q in modulating NSC cell behavior.

The complement system is an enzymatic cascade of proteins that plays a crucial role as the first line of defense against pathogens as it contributes to both innate and adaptive immune responses (*Dunkelberger and Song, 2010*). C1q is the recognition molecule of the classical pathway of complement activation. The traditional role of C1q in the immune system is to recognize and bind to antigen-antibody immune complexes, pathogens, lipids, and proteins accumulated during apoptosis or present on cell debris, initiating autocatalytic activation of the complement cascade and/or driving debris clearance by phagocytic immune populations. Recently, C1q has been found to act in non-traditional roles (*Peterson and Anderson, 2014*). In the CNS, C1q mediates the elimination of low activity presynaptic terminals by microglia (*Stephan et al., 2012*; *Presumey et al., 2017*) and modulates axon growth and guidance by masking myelin-associated glycoprotein-mediated growth inhibitory signaling (*Peterson et al., 2015*). In the muscle, C1q in C1 complex activates canonical Wnt signaling via conformation-induced activation of C1s serine protease activity, promoting age-associated decline in regeneration (*Naito et al., 2012*). In both cases, these activities are independent of complement cascade activation but remain consistent with the recognition functions of C1q in the immune system (*Botto et al., 1998*; *Mevorach et al., 1998*; *Nauta et al., 2002*). Additionally, however, C1q induces ERK signaling in fetal cytotrophoblasts (*Agostinis et al., 2010*), and binds discoidin domain receptor 1 (DDR1), directly activating mitogen-activated protein kinases and PI3K/Akt in hepatocellular tumor cells (*Lee et al., 2018*). This result suggests that C1q could play additional non-traditional roles, functioning as a ligand that can initiate cell signaling and/or directly interact with a transmembrane receptor to mediate cell signaling. Supporting this notion in CNS cells, C1q in the absence of other complement components induces neuronal-gene expression critical for cell survival in vitro (*Benoit and Tenner, 2011*), and our recent data link C1q to direct effects on NSC function and signaling (*Hooshmand et al., 2017*).

We hypothesize that there is receptor-mediated communication between the host immune system and endogenous and/or donor NSC in the injured, diseased, or aged CNS that is mediated by C1q. Consistent with such a role, C1q protein/mRNA increases and persists locally in the CNS after injury and in aging/disease as a result of disruption to the blood–brain barrier and local synthesis by invading and resident inflammatory cells (*Peterson and Anderson, 2014*; *Nguyen et al., 2008*; *Hooshmand et al., 2017*; *Dunkelberger and Song, 2010*; *Anderson et al., 2004*). In this study, we test whether C1q directly modulates NSC behavior through cell surface protein interactions and receptor-mediated cell signaling transduction mechanisms. We show for the first time that C1q activates specific intracellular signaling pathways in NSC to modulate NSC behavior. Using an unbiased screening strategy, we identify five novel transmembrane C1q signaling/receptor candidates in NSC, and define specific functions for the first of these, CD44, establishing the biological relevance of C1q-receptor interactions. These findings reveal an important role of complement C1q in modulating NSC behavior and identify a novel element in the understanding of the neuro-immune interface and NSC biology.

## Results

### C1q induces specific activation of intracellular signaling pathways on hNSC

We investigated the effects of C1q on intracellular signaling in hNSC by testing three physiological C1q concentrations that are produced by PMN [28 ng/mL $\cong$ 0.1 nM] (*Hooshmand et al., 2017*), Mφ [400 ng/mL $\cong$ 1.0 nM] (*Hooshmand et al., 2017*), or present in plasma. Reported levels of circulating C1q range between 40 and 120 µg/mL (*Dillon et al., 2009*; *Delamarche et al., 1988*; *Yonemasu et al., 1978*); in this study, we used [80 µg/mL $\cong$ 200 nM]. hNSC were treated with purified human C1q for 15, 30, and 60 min, and activation of intracellular signaling pathways assessed using a quantitative phosphoarray analysis that can simultaneously detect 18 phosphorylated proteins that are part of well-described intracellular signaling pathways (*Figure 1A* and *Figure 1—figure supplement 1*; a 1.5-fold increase was defined as the minimum threshold for positive response). Phosphoarray analysis demonstrated rapid activation of specific and selective signaling pathways

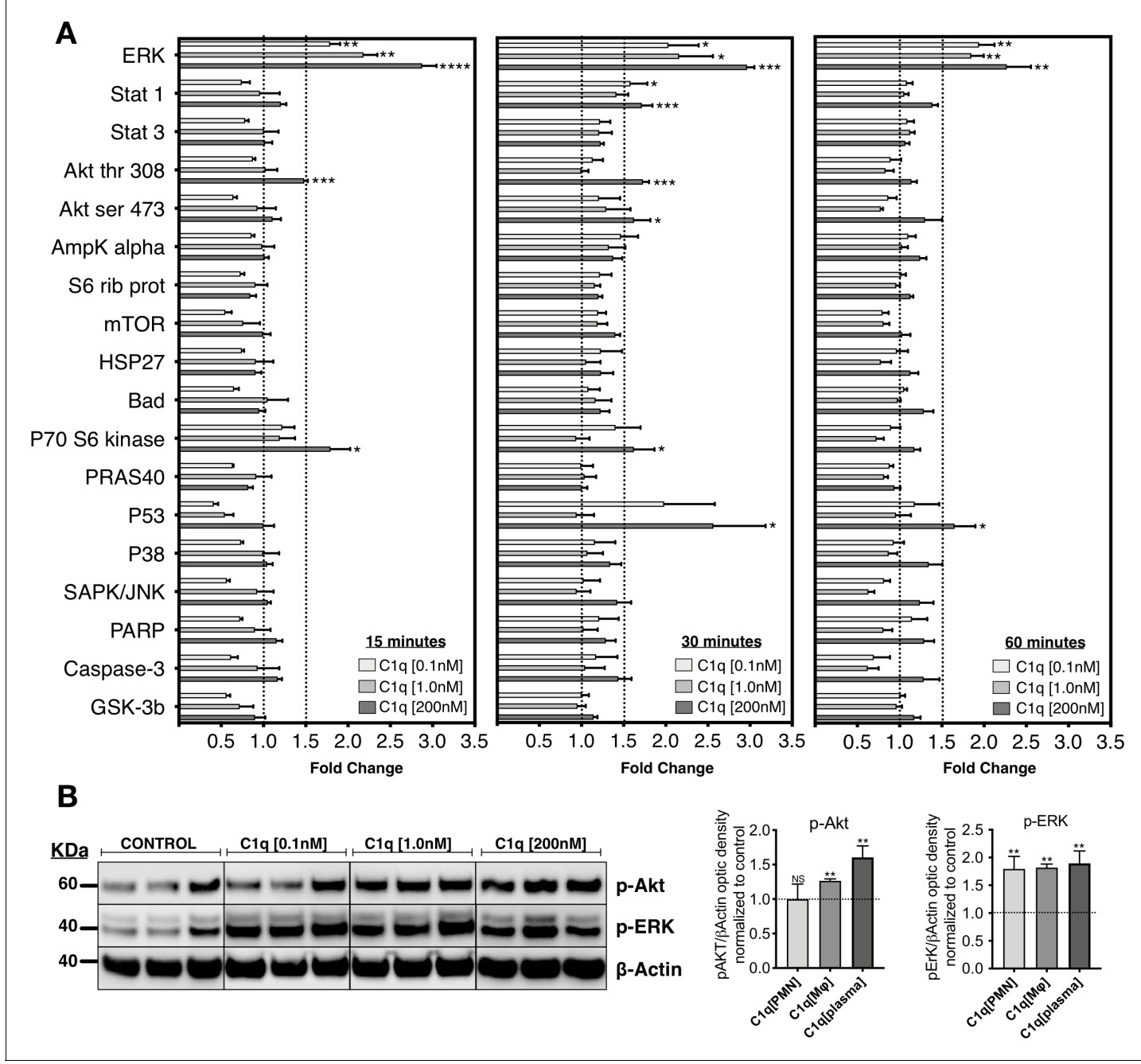

**Figure 1.** C1q induces intracellular signaling activation in hNSC. (**A**) Phosphoarray analysis quantification of hNSC exposed to C1q [0.1 nM], [1.0 nM], or [200 nM] concentrations for 15, 30, or 60 min. Data show mean ± SEM (N = 3 biological replicates per condition) for quantified optical intensity, normalized to control (dashed line = 1). In addition to statistical analysis, using one-sample t-test (*p≤0.05, **p≤0.01, ***p≤0.001), a minimum threshold of 1.5-fold activation (dashed line) was set for statistical significance. (**B**) ERK and Akt activation was verified using western blot in lysates from hNSC exposed to C1q [0.1 nM], [1.0 nM], or [200 nM] concentrations for 60 min. p-ERK and p-Akt band intensities were normalized to β-actin. Data show mean ± SEM (N = 3 biological replicates per condition) for quantified optical intensity normalized to control (dashed line). Statistical analysis using one-sample t-test (NS, not significant; *p≤0.05, **p≤0.01, ***p≤0.001).

The online version of this article includes the following figure supplement(s) for figure 1:

**Figure supplement 1.** C1q induces intracellular signaling activation in hNSC.

after C1q treatment, with MAPK/ERK signaling showing the most robust effect at all tested C1q concentrations. C1q at 200 nM also significantly and selectively activated PI3K signaling, identified by p-Akt and P70 S6 kinase, as well as P53. Phosphoarray results were confirmed using western blot for intracellular activation of p-Akt and p-ERK after 60 min of C1q exposure (*Figure 1B*). These data demonstrate rapid C1q-mediated activation of specific intracellular signaling pathways, consistent with a receptor-mediated mechanism.

## Unbiased screening strategy for cell surface C1q-binding partners on hNSC identifies five novel candidates

Based on the observation of C1q-induced intracellular signaling pathway activation, we followed an unbiased screening strategy to identify cell surface proteins on hNSC that interact with C1q. In a first approach, we used a far-western blot strategy to qualitatively establish the ability of C1q to interact with hNSC cell surface proteins (*Figure 2A*). We first verified whether C1q could bind a multitude of intracellular proteins in total protein lysates in this assay, consistent with the conventional role of C1q as a recognition molecule of the classical complement pathway. Next, we tested C1q binding, specifically, in the cell surface fraction. In contrast to the total protein fraction, we identified few bands, suggesting that C1q interacts specifically with a small number of cell surface binding partners on hNSC.

We sought to identify cell surface C1q binding partners using mass spectrometry nanoLC-MS/MS analysis of samples generated by two parallel strategies: 1) pull-down of whole cell hNSC protein isolates with biotinylated C1q immobilized on a streptavidin column; 2) pull-down of hNSC membrane protein fraction isolates generated after incubation and cross-linking of live hNSC with C1q. For the first approach, after validating that biotinylated and non-biotinylated C1q shared similar binding patterns (*Figure 2—figure supplement 1A–C*) and binding specificity (*Figure 2—figure supplement 1D,E*), total protein extract was incubated with immobilized C1q followed by a stringent wash protocol to minimize detection of nonspecific interactions. Bound proteins were eluted and analyzed using SDS-PAGE followed by silver staining (*Figure 2B*), and by nanoLC-MS/MS, which identified over 400 protein families. However, consistent with the far-western blot data and known interactions between C1q and intracellular constituents, few of these were known cell surface proteins.

For the second approach, we designed a novel strategy to selectively identify cell surface protein interactions. A critical limitation of both far-western blots and conventional pull-down assays is that these experiments are performed with denatured proteins, resulting in 3D structure loss. Our second approach targeted assessment of C1q-hNSC cell surface protein interaction in a biologically relevant setting where the 3D structure of ligand and candidate receptors is maintained. To accomplish this, cell surface interactions were initiated in live cells (*Figure 2C*). Monolayer hNSC were incubated with purified human C1q followed by cross-linking bound C1q to interacting proteins using a cell impermeable cross-linker (Sulfo EGS) with a spacer arm of 16.1 A° to cross-link only closely interacting proteins. Membrane proteins cross-linked with C1q were then extracted, pulled-down with biotinylated specific anti-C1q antibody, and bound proteins eluted and analyzed using nanoLC-MS/MS. This approach identified a short list of five known transmembrane proteins representing novel candidate C1q receptor and signaling protein interactions (*Figure 2D*): CD44, GPR62, BAI1, c-MET, and ADCY5. RT-PCR analysis confirmed positive expression of these five candidates in hNSC (*Figure 2D*). To confirm in vivo expression of the identified candidates, we evaluated published single-cell RNAseq data (*Figure 2E*) from the human embryonic cortex at gestational weeks' 22–23 (*Fan et al., 2018*). These data show expression in the NSC population during human development, as well as the emergence of cell type-specific expression patterns consistent with the published literature for these candidates (*Zhang et al., 2014*), for example, the expression of CD44 in maturing astrocytes (*Liu et al., 2004*).

C1q–signaling candidate protein interactions were validated using proximity ligation assay (PLA), in which positive fluorescence (red punctae) indicates direct binding (*Figure 3A*). A quantitative comparison of protein interaction was performed by analyzing the number of fluorescent punctae per cell. PLA validation was performed with a negative mis-matched ligand control C1q-C3aR (*Figure 3B,C*) and a positive matched ligand control C3a-C3aR (*Figure 3D,E*). *Figure 3F–O* show representative images of PLA detection of C1q-CD44, C1q-GPR62, C1q-BAI1, C1q-c-MET, and C1q-ADCY5 interaction, respectively, as well as quantification of PLA binding for each signaling/receptor

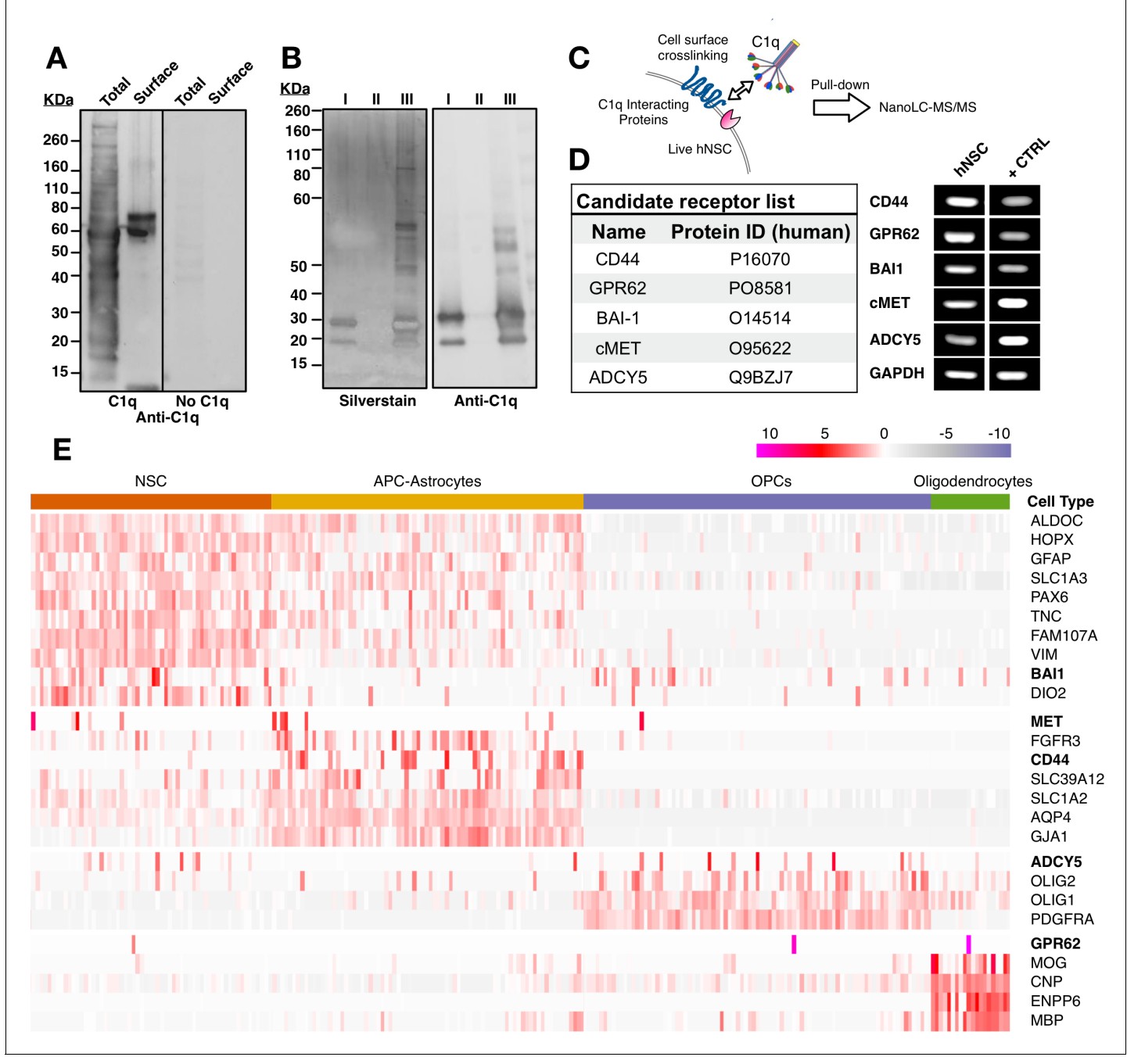

**Figure 2.** Unbiased screening for cell surface C1q-binding partners on hNSC identifies five novel signaling candidates. (**A**) C1q binds specifically to cell surface proteins in hNSC. Far-western blot reveals C1q binding interactions in hNSC protein fractions derived from either total (multiple bands) or cell surface (few bands) isolates. Exclusion of C1q (no C1q control) reveals no anti-C1q staining. (**B**) Silver stain (left) and anti-C1q western blot (right) analysis in C1q pull-downs with hNSC total protein lysates. Conditions: (I) column + C1q (bait) and no total protein lysate (prey), verifying C1q attachment to the column; (II) column + total protein lysate (prey) and no C1q (bait), confirming the absence of nonspecific protein binding to the column; (III) column + C1q (bait) + total protein lysate (prey), identifying multiple bands associated with C1q. (**C**) Experimental scheme for identification of C1q binding partners on hNSC cell surface using a cell surface crosslinking strategy followed by pull-down and mass spectrometry analysis. (**D**) Cell surface-C1q crosslinking using the strategy from (**C**) identifies five novel membrane-bound signaling candidates. RT-PCR analysis in hNSC and positive control samples (+CTRL, human universal RNA) demonstrates the positive expression of each target signaling candidate by hNSC. (**E**) Human embryonic cortex single-cell RNA-seq dataset analysis verifies C1q candidate receptor expression on NSC. Heatmap clustered on the gene expression of glia cell type identity gene markers and C1q candidate receptors gene reveals a subset of glia cells that express the candidate receptor genes. Glia cells were processed and categorized into NSC (radial glia), astrocyte precursor cells (APC)/astrocytes, oligoprogenitors (OPCs), and oligodendrocytes.

*Figure 2 continued on next page*

*Figure 2 continued*

The online version of this article includes the following figure supplement(s) for figure 2:

**Figure supplement 1.** Biochemical analysis of C1q and biotinylated-C1q.

candidate. These data identify and validate C1q interaction with identified receptor/signaling candidates on hNSC, all of which represent novel binding partners.

GPR62 and BAI1 are classical G-protein coupled receptors (GPCRs). Brain angiogenesis inhibitor protein 3 (BAI3) is a BAI1 homolog that has already been shown to bind to C1q and C1q like proteins in the CNS (*Bolliger et al., 2011*), supporting our findings from the candidate screening strategy. GPR62 is an orphan receptor that has no previously identified ligands (*Lee et al., 2001*). c-Met is a receptor tyrosine kinase (RTK) for which hepatocyte growth factor is the only known ligand (*Organ and Tsao, 2011*). ADCY5 belongs to the family of membrane-bound adenylate cyclases (AC) that modulate downstream signaling effectors such as cAMP-dependent protein kinase A (PKA)

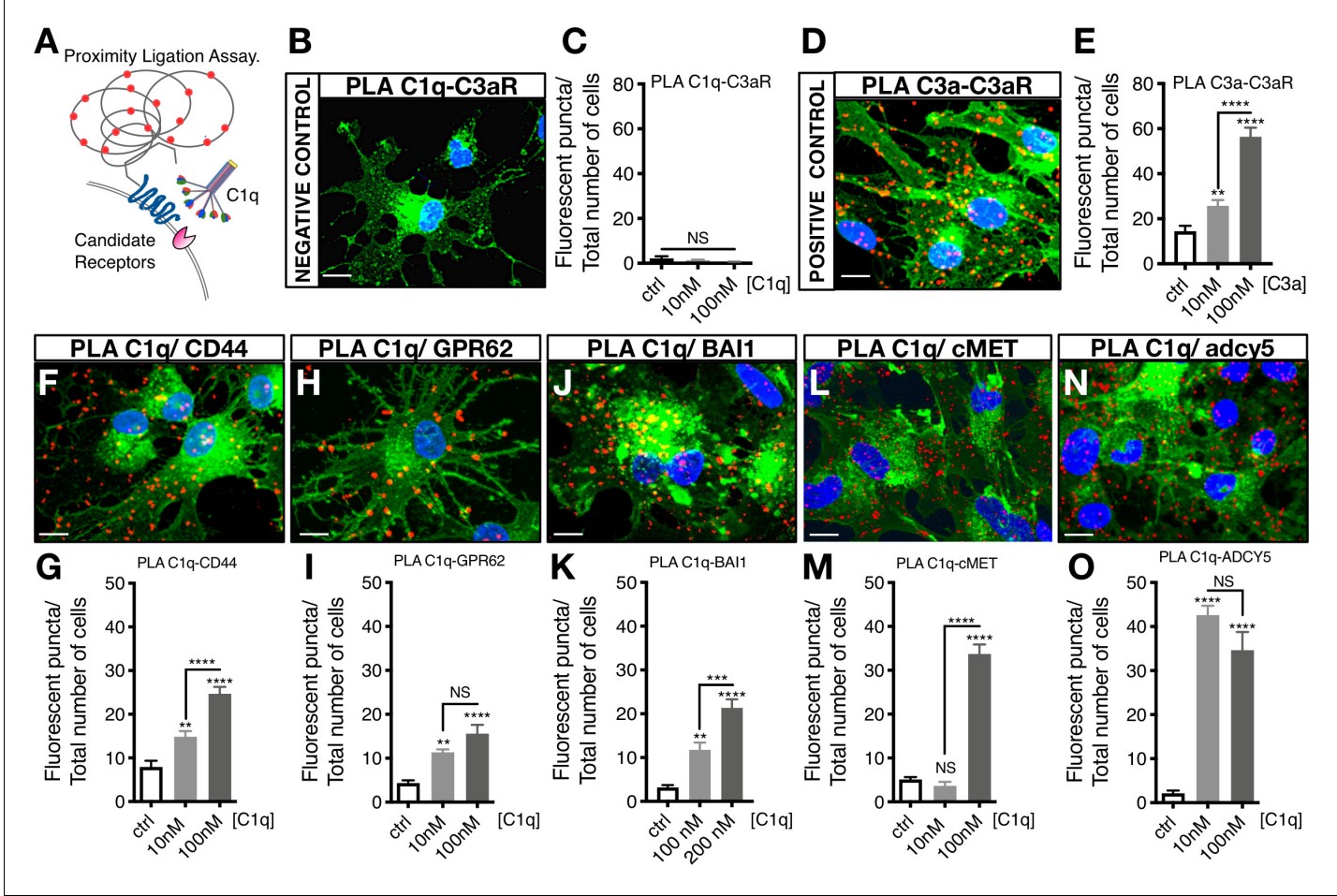

**Figure 3.** Proximity Ligation Assay (PLA) verifies C1q binding with five novel signaling candidates. (**A**) In proximity ligation assays quantifiable positive red fluorescent punctae are obtained only when two target proteins are interacting nearby (<30 nm). (**B**) PLA validation showing a representative image from a mis-matched ligand-receptor negative control (C1q-C3aR); quantification in (**C**). (**D**) PLA validation showing a representative image from a matched ligand-receptor positive control (C3a-C3aR); quantification in (**E**). PLA representative images and quantification of C1q with novel signaling candidates: (**F,G**) C1q-CD44; (**H,I**) C1q-GPR62; (**J,K**) C1q-BAI1; (**L,M**) C1q-c-MET; and (**N,O**) C1q-ADCY5. Membrane staining (green) shown with Alexa 488 conjugated wheat germ agglutinin (WGA). Scale bars 10 µm. Graphs represent quantification of the average number of fluorescent punctae/cell in 13 random pictures/experiment (n = 2 biological replicates; mean ± SEM). Statistical analysis of PLA punctae was significant via one-way ANOVA for all signaling candidates at p<0.05 and was followed by Tukey's post-hoc t-tests as indicated (NS, not significant; *p≤0.05, **p≤0.01, ***p≤0.001, ****p≤0.0001), no significant difference (ANOVA p=0.184) was identified in the negative control condition.

(*Sassone-Corsi, 2012*). Finally, CD44 is a multi-ligand, multifunctional transmembrane glycoprotein that controls cellular signaling through interactions with other cell surface receptors, including RTKs (e.g., c-Met) (*Orian-Rousseau et al., 2002*), GPCRs (e.g., CXCR4) (*Fuchs et al., 2013*), and Wnt-induced beta-catenin/LRP6 (*Schmitt et al., 2015*; *Orian-Rousseau and Schmitt, 2015*).

We hypothesized that selected C1q signaling/receptor candidates may have specific in vitro and in vivo functions; for example, mediation of C1q induced migration vs. direction of hNSC fate or proliferation. If different C1q signaling/receptor candidates mediate different functions, these functions could potentially be linked to the activation of specific intracellular signaling pathways in hNSC. We evaluated the functional significance of specific intracellular signaling pathways in C1q-induced chemotaxis and C1q-modulation of proliferation on hNSC using intracellular signaling pathway inhibitors focusing on: 1) pERK/MAPK inhibition because C1q induces p-ERK signaling at all tested concentrations (PD98059); 2) G-protein coupled receptor inhibition because two candidates are GPCRs (pertussis toxin, PTX); and 3) p38 MAPK inhibition as a negative control because we did not detect p38 signaling induction (SB20358). We focused on migration and proliferation to minimize the potential for toxicity due to long-term inhibitor treatment, and because we have previously established the effects of C1q administration on these hNSC responses (*Hooshmand et al., 2017*; *Figure 4—figure supplement 1*). We first verified that a 48 hr exposure to the tested inhibitors did not result in toxicity as assessed by total cell number (data not shown; one-way ANOVA p=0.6226). C1q activation of p-ERK was effectively blocked by PD98059 in western blot (*Figure 4A,B*), and C1q-induced migration in transwell assays was selectively inhibited only by p-ERK inhibition (*Figure 4C*). In contrast, the effect of C1q on hNSC proliferation in BrdU incorporation assays was both p-ERK and GPCR dependent but not affected by p38 inhibition (*Figure 4D,E*). These data indicate that different signaling pathways mediate distinct effects on hNSC, consistent with the hypothesis that different receptors mediate different cellular functions.

## CD44 mediates C1q induced p-ERK intracellular signaling activation and migration in NSC in vitro

We focused on CD44 as an initial target through which to test the physiological role of C1q interaction with the identified hNSC receptor/signaling candidates. Given that CD44 has identified roles in p-ERK signaling (*Herishanu et al., 2011*; *Kashyap et al., 2018*) and glial progenitor migration (*Piao et al., 2013*), we hypothesized that CD44 may mediate C1q-induced p-ERK activation and chemotaxis in hNSC. To test this hypothesis, we generated wildtype (WT) and knockout (KO) CD44 hNSC using CRISPR Cas9 gene editing (*Figure 5—figure supplements 1–2*), using an hNSC line generated at UCI (UCI161) for these studies. Stable selection of CD44 WT and CD44 KO hNSC populations was verified at two passages (2 weeks), four passages (4 weeks), and seven passages (7 weeks) after genetic modification using flow cytometry, (*Figure 5—figure supplement 1D–G*), western blot (*Figure 5A*), and cell surface immunocytochemistry (*Figure 5B*), respectively. Stable CRISPR-Cas9 modification and homozygous generation of CD44 WT/KO cell lines were validated by genotyping via Sanger sequencing (*Figure 5—figure supplement 2*). After CRISPR cas9 gene editing normal cellular karyotype was confirmed (*Figure 5—figure supplement 3A–B*). CD44 KO did not alter CD133+ stem cell proportion (*Figure 5—figure supplement 3C,D*) or baseline hNSC proliferation in comparison with WT cells (*Figure 5—figure supplement 3E,F*). Maintenance of WT and KO hNSC in differentiation media (DM) for 14 d in vitro (DIV) revealed no differences in GFAP+/βIII tubulin- astroglial lineage cells or in GFAP-/βIII tubulin+ neuronal lineage cells (*Figure 5—figure supplement 3K–S*). However, CD44 KO cells exhibited a small but significant increase in nuclear Olig2+ oligodendroglial lineage cells (KO 30% vs. WT 21%, p<0.0001; *Figure 5—figure supplement 3H–J*). These data demonstrate that CD44 KO cells retained normal baseline characteristics for hNSC, including multipotency.

We investigated the functional effect of CD44 KO on hNSC intracellular signaling in response to C1q. CD44 KO cells did not exhibit p-ERK activation in response to C1q [0.1 nM] or C1q [1 nM] in comparison with WT cells, and exhibited a significant attenuation in response to C1q [200 nM] (*Figure 5C,D*). Based on the effect of p-ERK inhibition on hNSC migration (*Figure 4F*), we subsequently tested whether CD44 mediates C1q-induced hNSC migration in transwell assays. Baseline migration in control media was not significantly different between WT and CD44 WT and KO hNSC (Student's t-test, p=0.1785). As shown earlier (*Hooshmand et al., 2017*) and in *Figure 4—figure supplement 1D*, CD44 WT hNSC exhibited concentration-dependent migration in response to C1q

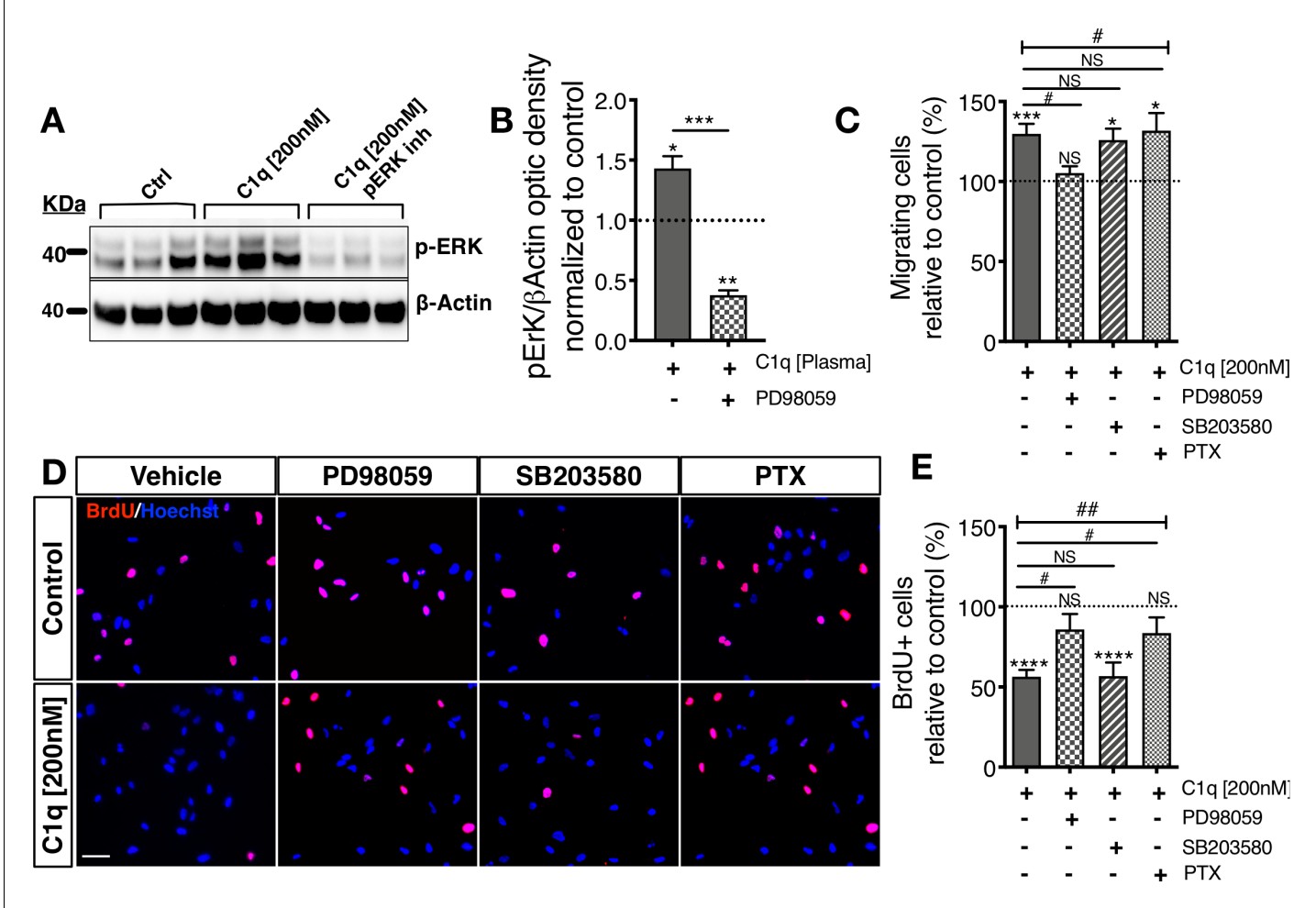

**Figure 4.** Distinct cellular signaling pathways control different C1q induced hNSC behavior. (**A,B**) Western blot analysis in cells treated with the MEK/ERK pathway inhibitor PD89059 verifies effective p-ERK inhibition in cells exposed to C1q [200 nM] for 60 min. Optical densities were normalized to untreated controls (dashed line). Statistical analysis using one-sample t-test (*p≤0.05, **p≤0.01) for comparison with control and Student's t-test for comparison between conditions as indicated (***p≤0.001). N = 3 biological replicates. (**C**) C1q [200 nM] induces hNSC chemotaxis in transwell migration assays (shown also in **Figure 4—figure supplement 1D**), which is blocked by PD89059 inhibition of p-ERK. No effect was observed with either a GPCR inhibitor (Pertussis toxin; PTX) or p38 MAPK inhibitor (SB203580). (**D,E**) C1q [200 nM] decreases hNSC proliferation in vitro as analyzed by BrdU incorporation (shown also in **Figure 4—figure supplement 1E–F**), which is blocked by either PD89059 or PTX treatment. No effect was observed with the p38 inhibitor SB203580. (**D**) Representative pictures of BrdU+ cells in control hNSC or hNSC treated with C1q [200 nM] and pathway inhibitors as indicated. (**E**) Brdu+ nuclei quantification (%). Data shows mean ± SEM normalized to untreated controls (dashed line). Statistical analysis using one-sample t-test (NS, not significant; *p≤0.05, **p≤0.01, ***p≤0.001) for comparison with untreated controls, and using one-way ANOVA (#) p<0.05 followed by Tukey's post-hoc t-tests as indicated (NS, not significant; #p≤0.05) for comparison between conditions, N = 4 biological replicates. The online version of this article includes the following figure supplement(s) for figure 4:

**Figure supplement 1.** C1q induces NSC motility and chemotaxis in vitro.

[0.1 nM] and [200 nM] (**Figure 5E**). By contrast, CD44 KO hNSC did not migrate in response to C1q [0.1 nM], and migration was dramatically attenuated in response to C1q [200 nM] (**Figure 5E**, N = 4), demonstrating that CD44 is involved in C1q-induced signaling and chemotaxis in hNSC. The lack of complete migration blockade in response to C1q [200 nM] could be related to concentration-dependent activation of ERK/MAPK pathway signaling via C1q interaction with other receptor candidates.

To validate that these results were not due to off-target effects of CRISPR-Cas9, we employed a parallel genetic approach, generating mouse NSC (mNSC) from either CD44 WT or CD44 KO littermate mouse embryos (**Figure 5—figure supplement 4A–D**). We confirmed that mNSC exhibit the expression of CD44 and the other C1q signaling/receptor candidates (**Figure 5—figure**

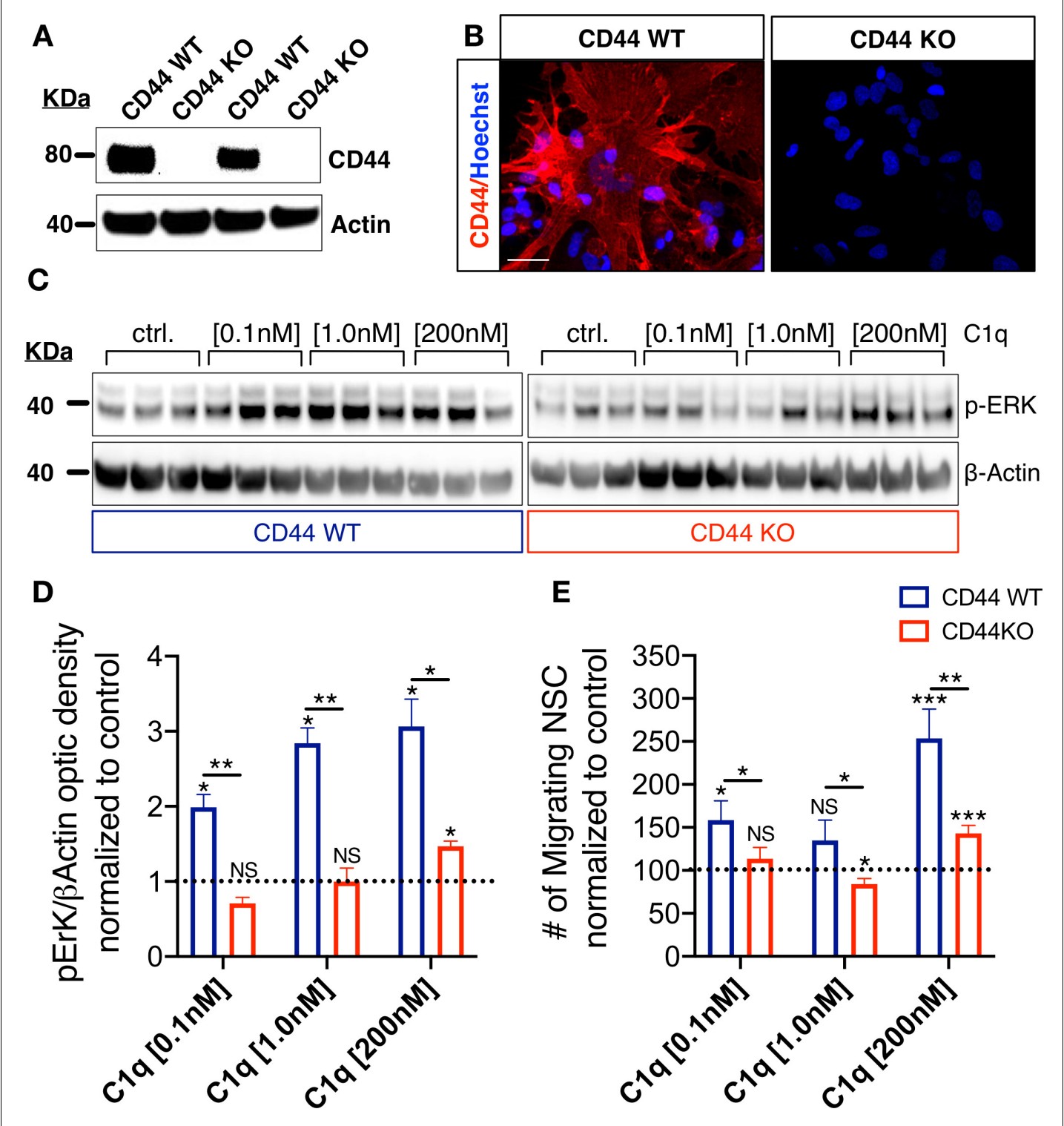

**Figure 5.** CD44 mediates C1q-induced ERK signaling activation and chemotaxis in hNSC in vitro. (**A,B**) CD44 expression profile in WT but not in CD44 KO hNSC was verified by (A) western blot and (B) Immunocytochemistry (scale bar 30 μm). (**C,D**) p-ERK western blot analysis of protein lysates from WT and CD44 KO hNSC shows that C1q-induced ERK activation is completely blocked in CD44 KO hNSC at C1q [0.1 nM] and [1 nM], and dramatically attenuated at C1q [200 nM]. p-ERK band optical intensities were normalized to β-actin within condition. Data show mean ± SEM (N = 3 biological replicates per condition) normalized to untreated controls for comparison between conditions (dashed line). Statistical analysis using one-sample t-test (NS, not significant; *p≤0.05) for comparison with control and Student's t-test for comparison between WT and CD44 KO hNSC as indicated (*p≤0.05,

*Figure 5 continued on next page*

Figure 5 continued

**p≤0.01). (E) Transwell chemotaxis assay reveals that C1q-induced hNSC chemotaxis is completely blocked in CD44 KO hNSC at C1q [0.1 nM] and dramatically attenuated at C1q [200 nM]. Data show mean ± SEM (N = 3 biological replicates per condition) normalized to untreated controls (dashed line). Statistical analysis using one-sample t-test (NS, not significant; *p≤0.05) for comparison with control and Student's t-test for comparison between WT and CD44 KO hNSC as indicated (*p≤0.05, **p≤0.01).

The online version of this article includes the following figure supplement(s) for figure 5:

**Figure supplement 1.** WT and CD44 KO hNSC generation.
**Figure supplement 2.** CD44 WT and CD44 KO hNSC genotyping by sequencing.
**Figure supplement 3.** Karyotype, stemness, proliferation capacity, and multipotency are unaltered after CRISPR Cas9 genetic deletion of CD44 in hNSC.
**Figure supplement 4.** CD44 WT and KO mNSC cell line generation.

supplement 4E), as well as a comparable response to complement components and the inflammatory injury microenvironment (*Hooshmand et al., 2017*), including induction of mNSC chemotaxis (*Figure 5—figure supplement 4F*). These data demonstrate the conservation of this mechanism across species. The loss of CD44 protein expression in KO vs. WT mNSC was confirmed using immunocytochemistry (*Figure 5—figure supplement 4D*), and transwell assays were used to verify C1q induced migration as for hNSC. Baseline migration in control media was identical between CD44 WT and KO mNSC (Student's t-test, p=0.6150). However, while WT mNSC exhibited increased migration in response to C1q (*Figure 5—figure supplement 4F,G*) similar to WT hNSC (*Figure 4—figure supplement 1D*, *Figure 4E*), this response was completely blocked in CD44 KO mNSC (*Figure 5—figure supplement 4G*). Indeed, CD44 KO mNSC exhibited a repulsive effect in response to C1q. Temperature inactivation of human C1q abolished all responses (*Figure 5—figure supplement 4G*), confirming both specificity and a requirement for functional 3D conformation of the C1q protein. In summary, these in vitro findings demonstrate that CD44 is involved in C1q-mediated signaling pathway activation and chemotaxis in mammalian NSC.

## C1q modulation of hNSC proliferation and differentiation is CD44 independent

We have previously shown that C1q alters hNSC proliferation and differentiation (*Hooshmand et al., 2017*). We therefore tested whether C1q acts through CD44 to mediate these effects in addition to cell migration. We assessed cell proliferation via EdU incorporation after 2DIV in DM in the presence or absence of C1q. No difference was observed between CD44 WT and KO hNSC baseline proliferation (*Figure 5—figure supplement 3E–F*, noted above), or proliferation response to C1q at any tested C1q concentration (ANOVA and Tukey post-hoc t-tests as indicated in *Figure 6A–C*; Tukey p=0.999). These data demonstrate that the effects of C1q on hNSC proliferation are CD44 independent.

Next, we tested the effect of C1q on the fate of CD44 WT and KO hNSC during 14DIV differentiation in DM, analyzing nuclear Olig2, GFAP, and βIII-tubulin immunolabeling. When exposed to C1q at [0.1 nM] or [1.0 nM], WT hNSC exhibited a significant increase in Olig2+ cells vs. control (*Figure 6D–F*; C1q [0.1 nM] 29% vs. control 21%, Tukey p=0.022; C1q [1.0 nM] 28% vs. control 21%, Tukey p=0.048), which was attenuated in CD44 KO hNSC (*Figure 5E–F*; C1q [0.1 nM] 29% vs. control 30%, Tukey p=0.999; C1q [1.0 nM] 31% vs. control 30%, Tukey p=0.999). This result is likely to reflect an indirect effect of increased Olig2+ cell number observed under baseline differentiation conditions for CD44 KO vs. WT hNSC (30% vs. 21%; *Figure 5—figure supplement 3G,H*, noted above), creating a ceiling effect for C1q-driven increases in Olig2+ cell number. Byn contrast, when exposed to C1q [200 nM] both WT and CD44 KO hNSC exhibited a significant and equivalent decrease in Olig2+ cells (*Figure 6F*; in CD44 WT: C1q [200 nM] vs control 10% vs. 21%, Tukey p<0.0001; KO C1q [200 nM] vs control 14% vs. 30%, Tukey p<0.0001). CD44 WT and KO hNSC exhibited comparable concentration-dependent responses to C1q in astroglial (GFAP+/βIII-tubulin-, *Figure 6G–I*), neuronal (GFAP-/βIII-tubulin+, *Figure 6J–L*), and undecided (GFAP+/βIII-tubulin+ cells, *Figure 6M–O*) cell fate, with no significant differences observed between CD44 WT and KO hNSC in response to C1q at any tested C1q concentration (ANOVA and Tukey post-hoc t-tests as indicated in *Figure 6*). In accordance with this observation and as noted above, no significant differences were observed between CD44 WT and CD44 KO hNSC baseline astroglial or neuronal fate

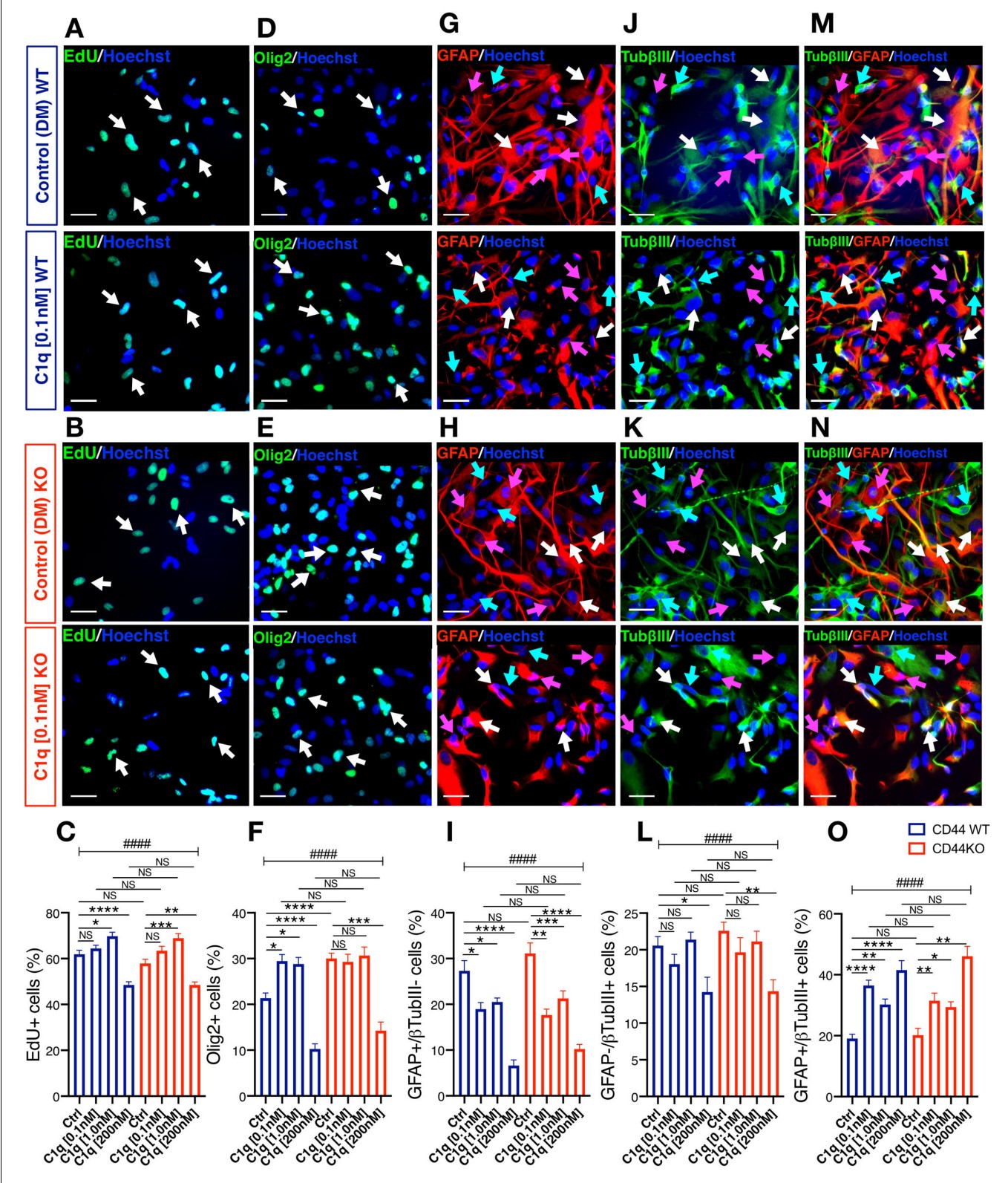

**Figure 6.** C1q modulation of hNSC proliferation and fate is CD44 independent. (**A-C**) EdU incorporation analysis reveals that C1q exposure for 2DIV (0.1 nM–200 nM vs. control media, as indicated) exerts comparable effects on hNSC proliferation in CD44 WT and KO. Representative images (A, B; white arrows, scale bar 30 μm) and quantification (**C**) of EdU+ nuclei of CD44 WT (blue) and KO (red) hNSC. (**D-O**) Immunohistochemical fate analysis reveals that C1q exposure for 14DIV in DM (0.1 nM–200 nM vs. control media, as indicated) exerts comparable effects on hNSC fate in CD44 WT and

*Figure 6 continued on next page*

Figure 6 continued

KO. hNSC were immunolabeled for Olig2 (green) to identify oligodendroglial lineage cells, and doubled immunolabeled for glial fibrillary acidic protein (GFAP, red) and Tubulin βIII (TubβIII, green) to identify astroglial and neuronal lineage cells, respectively. Representative images and quantification of nuclear Olig2+ oligodendroglial cells (D-F; white arrows, scale bar 30 µm), GFAP+/TubβIII- astroglial cells (G-I; pink arrows, scale bar 30 µm), GFAP-/TubβIII+ neuronal cells (j-l, green arrows, scale bar 30 µm), and GFAP+/TubβIII+ double-positive 'undecided' cells (M-O, white arrows, scale bar 30 µm). Data represent average number of cells as a percentage of total nuclei ± SEM of EdU labeled or immunolabeled cells obtained from 10 random fields/experiment (N = 4 biological replicates for each label and condition; average 140 ± 60 cells/field). Statistical analysis was performed using one-way ANOVA (####$p \leq 0.0001$) followed by Tukey's post-hoc t-tests as indicated (NS, not significant; *$p \leq 0.05$, **$p \leq 0.01$, ***$p \leq 0.001$, ****$p \leq 0.0001$). No significant differences were observed between WT and CD44 KO hNSC in response to C1q at any tested C1q concentration (Tukey post-hoc t-tests $p \geq 0.7891$ for all comparisons).

(*Figure 5—figure supplement 3K–S*). In all, these data are consistent with our previous report of a dose dependent effect of C1q on hNSC differentiation (*Hooshmand et al., 2017*), and demonstrate that the effects of C1q on astroglial and neuronal fate are independent of CD44. These data suggest that C1q functions through one or more of the other receptor candidates identified and their associated signaling pathways (e.g., GPCR signaling) to regulate hNSC proliferation and differentiation/fate.

CD44 KO decreases hNSC localization at the SCI epicenter, modulating astroglial fate and rescuing the capacity of hNSC to promote locomotor recovery after transplantation into the acute SCI microenvironment.

We have previously shown that the time of hNSC transplantation into the parenchyma adjacent to the SCI epicenter is a key variable for the in vivo fate and migration profile of engrafted donor human cells. Delayed hNSC transplantation (9–30/60 d post-SCI) results in extensive donor human cell migration away from the SCI epicenter, toward the distal rostral and caudal spinal cord, and predominant oligodendroglial differentiation (*Yonemasu et al., 1978*; *Salazar et al., 2010*; *Hooshmand et al., 2009*; *Cummings et al., 2006*). Conversely, acute hNSC transplantation (0 d post-SCI) results in increased donor human cell migration toward the SCI epicenter and a pronounced shift toward astroglial fate in the donor human cell population that engrafts adjacent to it (*Nguyen et al., 2017*; *Hooshmand et al., 2017*). Moreover, while delayed hNSC transplantation leads to locomotor recovery, acute transplantation does not. These effects can be rescued by specific systemic immunodepletion of the PMN cell population with anti-Ly6G antibody (*Nguyen et al., 2017*), as well as by C1q+C3a blockade via epicenter administration of neutralizing antibodies (*Hooshmand et al., 2017*). Based on these data and the identification of a novel role for CD44 expressed by hNSC in C1q-induced chemotaxis, we hypothesized that: 1) CD44 KO hNSC would exhibit reduced migration to the SCI epicenter after acute transplantation; 2) CD44 KO hNSC would exhibit reduced astroglial differentiation at the SCI epicenter after acute transplantation; and 3) CD44 KO hNSC would promote functional locomotor recovery after acute transplantation, while CD44 WT cells would not.

We first sought to demonstrate that there is a spatial and temporal C1q gradient, with highest levels in the injury epicenter and a peak in the acute period that could drive donor hNSC migration via C1q-CD44 interaction. We analyzed C1q protein in Rag one mice that received a T9 laminectomy alone (sham) or a T9 laminectomy plus a 50 kD contusion SCI, collecting spinal cord tissue in rostral, epicenter and caudal 3 mm segments (*Figure 7—figure supplement 1A*) at 3 hr, 24 hr or 9 d post-injury (pi). Using SDS-PAGE and western blot analysis, we verified that C1q protein is significantly increased at acute time points (3-24hpi) in the injury epicenter (*Figure 7—figure supplement 1A–C*). Plasma C1q circulates in the C1 complex, which is composed of C1q, C1r, and C1s. Also, a multitude of cells in the CNS can synthesize C1q, including PMN and Mφ, which do so both in vitro and in vivo after SCI (*Hooshmand et al., 2017*). Detection of C1q by western blot could thus indicate the presence of either C1q as a single protein or C1 complex. Consistent with this suggestion, microarray gene expression data overlapping these regions and time points shows a parallel increase in local C1q mRNA at the injury epicenter that is not accompanied by C1s mRNA (*Figure 7—figure supplement 1D,E*; *Aimone et al., 2004*), suggesting that locally synthesized C1q would be bioavailable for interaction as a single molecule, separate from C1 complex. Together, these data demonstrate that C1q is present in the SCI microenvironment acutely at the lesion epicenter, and supports

that C1q would be available to interact with CD44 and drive hNSC migration to the SCI epicenter after acute transplantation of donor hNSC.

To test if CD44 mediates hNSC recruitment toward the SCI epicenter after acute transplantation, 75,000 CD44 WT or KO hNSC were transplanted bilaterally into four sites (250 nL/site) in the intact parenchyma, two sites 1 mm rostral and two sites 1 mm caudal to the SCI epicenter, immediately after a moderate (50 kD) T9 contusion injury (*Nguyen et al., 2017*; *Hooshmand et al., 2017*). To enable adequate xenograft survival for analysis of survival and migration, we employed immunode-ficient Rag-1 mice; while the adaptive T and B cell response is absent in these animals, innate immune responses, including PMN infiltration, macrophage/monocyte infiltration, and early comple-ment molecules are retained. Donor CD44 WT and CD44 KO hNSC engraftment were observed in 100% of transplanted animals (N = 10/group). A subset of six animals was randomly selected from each group for immunohistology and unbiased stereological analysis (MBF StereoInvestigator). Fibronectin+ lesion volume was quantified using the Cavalieri probe. Similarly, the number of human-specific cytoplasmic marker (STEM121) and human-specific astrocyte marker (STEM123) posi-tive hNSC was quantified using the optical fractionator probe. Additionally, we have previously reported that, while hNSC in this acute transplantation paradigm do not invade the Fibronectin+ injury site per se, the robust magnitude of hNSC localized to the spared parenchyma adjacent to the SCI epicenter leads to undersampling in stereology; accordingly, we also analyzed human donor cell localization in this region using volumetric sampling with the Cavalieri probe (clustering) (*Nguyen et al., 2017*). All histological analyses were performed by observers blinded to the study groups 16 weeks post-transplant (WPT).

No differences in fibronectin+ lesion volume (*Figure 7—figure supplement 2A,B*), or in the num-ber of surviving engrafted donor human cells (*Figure 7—figure supplement 2C,D*; hNSC STEM121 + cells in brown) were observed in mice receiving CD44 WT vs. KO hNSC. However, consistent with our hypotheses, mice transplanted with CD44 KO hNSC exhibited a significant reduction in total STEM121+ donor cells adjacent to the SCI epicenter (*Figure 7A–C*). Similarly, mice transplanted with CD44 KO hNSC exhibited a significant reduction in hNSC derived STEM123+ astrocyte cluster-ing (h-astrocytes, *Figure 7D–H*) adjacent to the SCI epicenter and STEM123+ h-astrocytes in the spared parenchyma within 1 mm rostral-caudal to the SCI epicenter (*Figure 7I*). These data suggest that CD44 KO decreased the migration of transplanted hNSC toward the SCI epicenter, which was associated with decreased astroglial fate, and that this effect did not result from alterations in lesion volume or total donor cell engraftment and survival. Further, analysis of locomotor recovery on the horizontal ladder beam task, a sensitive task for animals exhibiting frequent stepping, showed that mice receiving CD44 KO hNSC made significantly fewer errors vs. mice receiving vehicle controls (*Figure 7J*), while no improvements were observed after transplantation of WT hNSC. Taken together, these data demonstrate that CD44 modulates hNSC migration and distribution in the in vivo acute microenvironment after SCI, affecting both lineage selection and the potential to yield locomotor recovery. While identification of C1q as a novel CD44 ligand suggests a novel mechanism by which hNSC CD44 expression may regulate the behavior of these cells in vivo, this experiment cannot rule out the potential contribution of other CD44 ligands in the SCI microenvironment.

## Epicenter C1q neutralization is comparable to CD44 KO in modulating hNSC behavior and capacity for repair after acute transplantation

CD44 has multiple ligands, including hyaluronic acid and components of the extracellular matrix (laminin, collagen , osteopontin; *Ponta et al., 2003*; *Weber et al., 1996*) many of which are altered in the injured spinal cord (*Struve et al., 2005*; *Hashimoto et al., 2003*; *Bernstein et al., 1985*). If C1q interaction is necessary for the role of CD44 in hNSC migration and capacity for repair, then we reasoned that C1q depletion at the SCI epicenter would be sufficient to reproduce the effect of genetic CD44 KO on transplanted hNSC. We employed a previously published approach and reagents (*Hooshmand et al., 2017*), transplanting WT hNSC into the rostral and caudal spared parenchyma immediately after T9 moderate contusion SCI, as described earlier. Acute hNSC trans-plantation was followed immediately by a single epicenter injection of either vehicle (hNSC + vehi-cle) or C1q neutralizing antibody (hNSC + C1qNAb). We have previously shown that there are no differences between NAb treatment, IgG control, and vehicle treatment with regard to hNSC cell engraftment and lesion volume (one-way ANOVA, p=0.493) (*Hooshmand et al., 2017*). Histological quantification was performed by blinded unbiased stereological analysis as described earlier, and

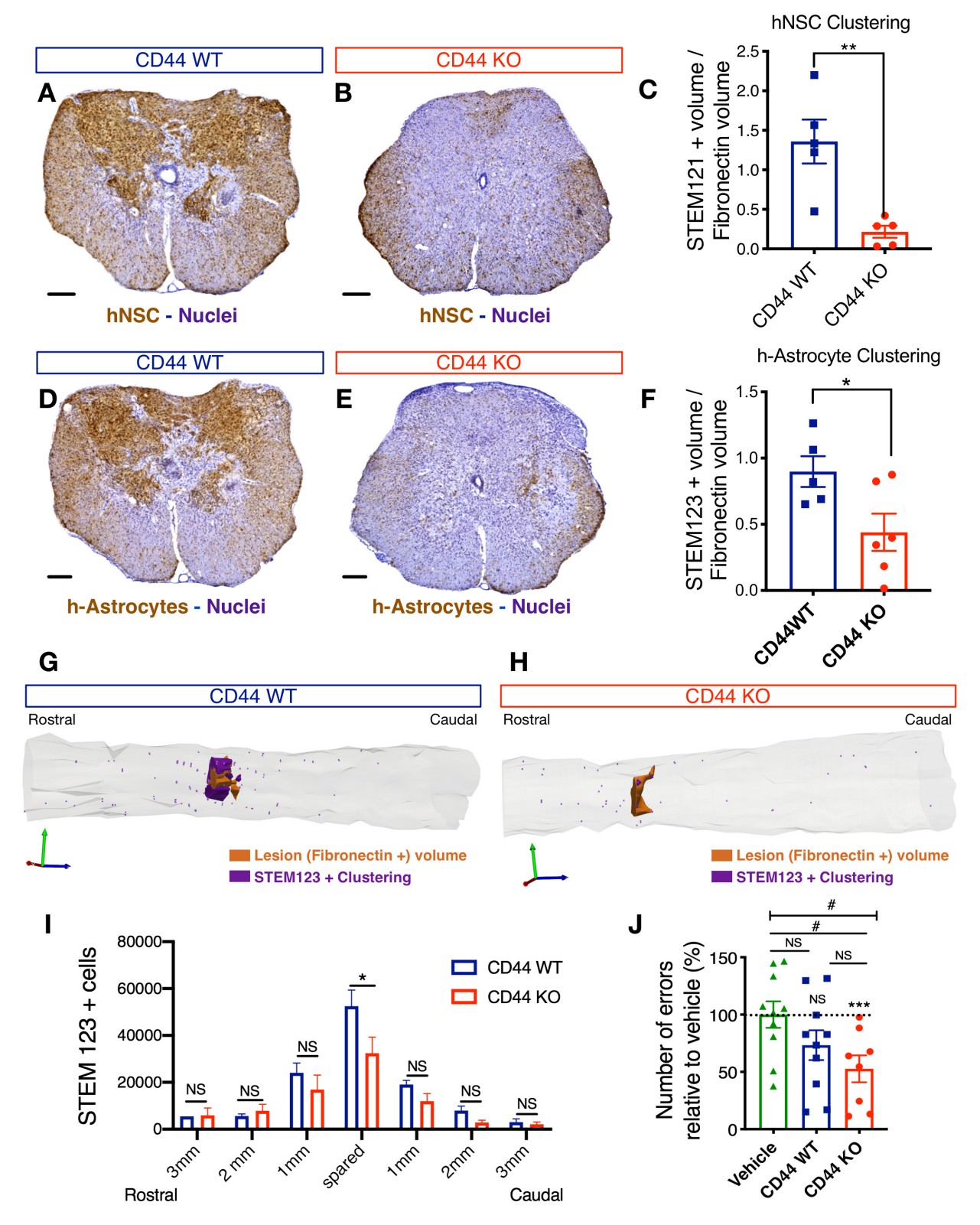

**Figure 7.** CD44 KO decreases hNSC localization at the SCI epicenter, modulating astroglial fate and rescuing the capacity of hNSC to promote locomotor recovery after transplantation into the acute SCI microenvironment. All histological data obtained 16 weeks post-transplant (WPT). (**A,B**) Representative images of transverse spinal cord sections corresponding to the injury epicenter area immunostained for the human-specific cell marker STEM121 (brown; scale bars, 250 μm) and hematoxylin nuclear counterstain (purple) on spinal cords from mice transplanted with WT (blue) or CD44 KO

*Figure 7 continued on next page*

*Figure 7 continued*

(red) hNSC. (C) Stereological analysis of STEM121+ human cell clustering using Cavalieri sampling demonstrates a decrease in CD44 KO vs. WT human cells adjacent to the SCI epicenter. Data show mean ± SEM (N = 5–6 mice randomly selected for histology/group). Statistical analysis using Student's t-test as indicated (**$p \leq 0.01$). (D,E) Representative image of transverse spinal cord section corresponding to the injury epicenter area immunostained for the human astrocyte specific marker STEM123 (brown; scale bars, 250 μm) and hematoxylin nuclear counterstain (purple) in spinal cords from mice transplanted with (D) WT or (E) CD44 KO hNSC. (F) Stereological analysis of STEM123+ h-astrocyte clustering using Cavalieri sampling demonstrates a decrease in CD44 KO vs. WT h-astrocytes adjacent to the SCI epicenter. Data show mean ± SEM (N = 5–6 mice randomly selected for histology/group). Statistical analysis using Student's t-test as indicated (*$p \leq 0.05$). (G,H) Representative three-dimensional reconstructions generated by stereological quantification of the lesion Fibronectin+ area (orange volume), STEM123+ clusters (purple volume), and individual STEM123+ cells (light purple dots) from mice transplanted with CD44 WT or KO hNSC. (I) Stereological analysis of STEM123+ h-Astrocyte number in the spared tissue adjacent to the SCI epicenter demonstrates a decrease in CD44 KO vs. WT localized in these regions. Data show mean ± SEM (N = 5–6 mice randomly selected for histology/group). Statistical analysis using Student's t-test as indicated (*$p \leq 0.05$). (J) CD44 KO in hNSC rescues the capacity of acutely transplanted hNSC to promote locomotor recovery. Mice transplanted acutely with CD44 KO but not WT hNSC made significantly fewer errors on the horizontal ladder beam locomotor task in comparison with vehicle-treated mice 16 WPT. Data shown as number of errors normalized to vehicle-treated mice (dashed line; N = 8–10 mice/group). Statistical analysis using one-sample t-test (NS, not significant, ***$p \leq 0.0005$) for comparison with control vehicle-treated mice and via one-way ANOVA (#) $p < 0.05$ followed by Tukey's post-hoc t-test as indicated (NS, not significant; #$p \leq 0.05$) for comparison between conditions.

The online version of this article includes the following figure supplement(s) for figure 7:

**Figure supplement 1.** C1q protein increases in the epicenter acutely after SCI.

**Figure supplement 2.** CD44 KO hNSC does not alter lesion volume or total hNSC engraftment after transplantation in the SCI acute microenvironment.

the effect of C1qNAb administration on lesion volume as well as hNSC engraftment and h-astrocyte localization analyzed at 12 WPT. As for CD44 WT vs. KO hNSC transplantation, no differences in fibronectin+ lesion volume (*Figure 8—figure supplement 1A,B*) or total donor STEM 121+ cell hNSC engraftment (*Figure 7—figure supplement 1C,D*) were observed in C1qNAb treated vs. vehicle-treated control mice. Similarly, consistent with the effect of CD44 KO, the number of hNSC that had committed to an astroglial fate adjacent to the SCI epicenter was significantly decreased in C1qNAb treated vs. vehicle-treated control mice at 12WPT (*Figure 8A–C*). Further, C1q NAb treatment also rescued the capacity of donor hNSC to promote recovery of locomotor function. While no differences in ladder beam errors were observed at 12 WPT (one-way ANOVA, p=0.4743; data not shown), the more sensitive CatWalk kinematic gait analysis showed significant improvement in mice receiving hNSC + C1qNAb for hindlimb duty cycle, hindpaw stand time, and support three paws, indicating increased weight- bearing vs. vehicle in a one-sample t-test (*Figure 8D–F*). Taken together, these data suggest that C1q blockade results in modulation of hNSC behavior and the potential of hNSC for repair in a manner that is consistent with CD44 KO in these cells.

Collectively, these data identify a novel ligand-receptor relationship for C1q and CD44, a novel role for CD44 in donor hNSC behavior, and a link between C1q-CD44 regulated donor hNSC distribution, fate and recovery of function after transplantation. Importantly, these findings suggest the potential for the optimization of cell therapies by manipulating hNSC responses to the host inflammatory microenvironment for CNS injuries and/or diseases.

## Discussion

C1q first appeared in the phylogenetic scale about 450 million years ago, and multiple studies suggest that this protein has evolved roles that are not restricted its functions in the innate immune response, particularly in the CNS (*Peterson and Anderson, 2014*). Here, we hypothesized that C1q signals to NSC as a functional ligand via receptor-mediated interactions to modulate stem cell behavior. Consistent with this hypothesis, we demonstrate the first evidence for a receptor-mediated signaling role for complement C1q in the CNS, modulating NSC migration, proliferation, and lineage commitment. Moreover, using an unbiased forward screen, we define and validate five cell surface signaling/receptor candidate proteins, all of which represent novel C1q binding partners. The novelty of this technique allowed us to identify cell surface binding partners for C1q while maintaining the 3D structure of the receptor and the ligand on the cell surface of live hNSC, and validated these biologically-relevant interactions with an optimized proximity ligation assay.

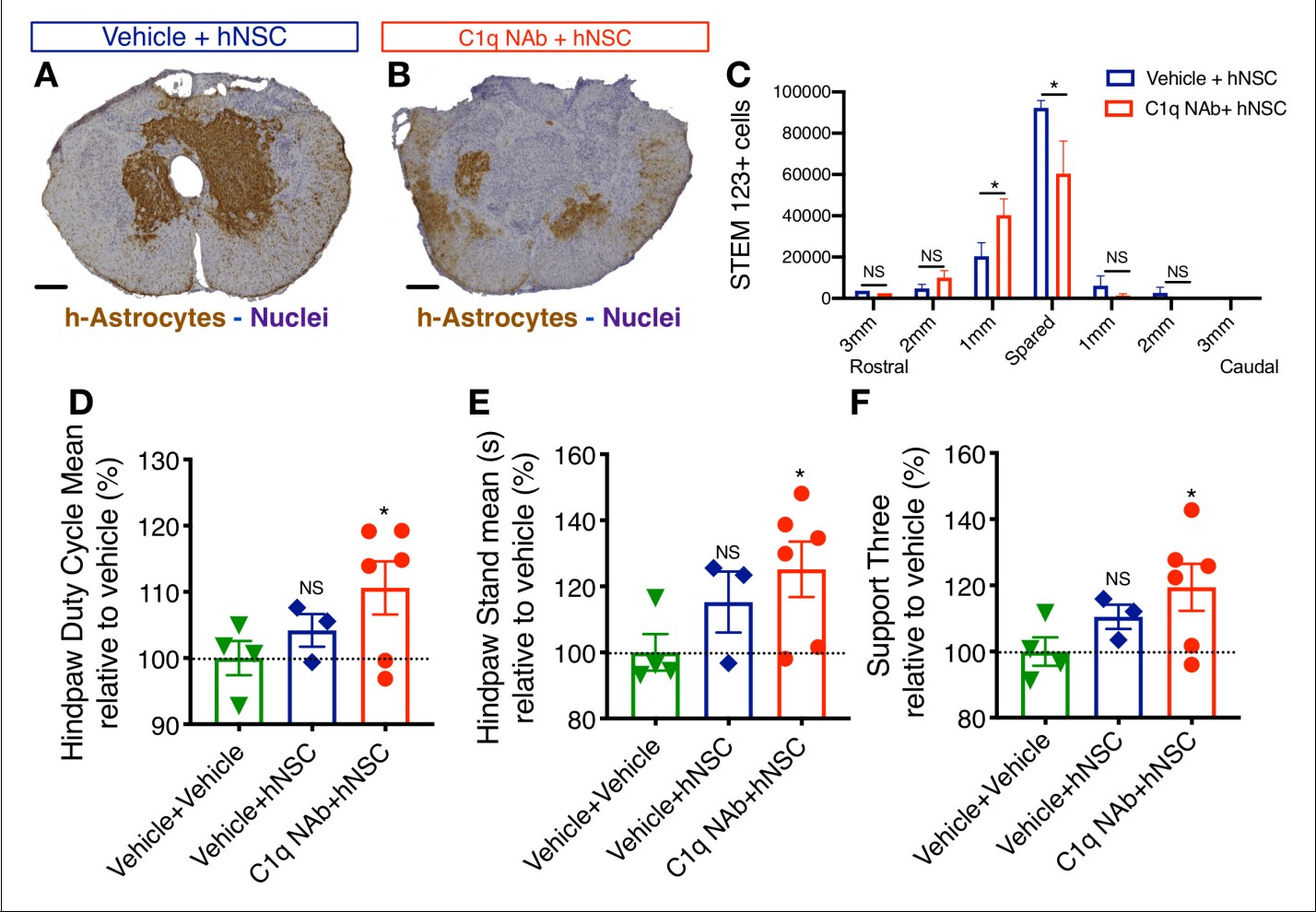

**Figure 8.** Epicenter C1q neutralization is similar to CD44 KO in modulating hNSC behavior and capacity for repair after acute transplantation. All histological data are from animals sacrificed at 12 weeks post-transplant (WPT). (**A,B**) Representative images of transverse spinal cord sections corresponding to the injury epicenter area immunostained for the h-Astrocyte STEM123 marker (brown, hematoxylin nuclear counterstain in purple; scale bars, 250 µm). (**A**) In the absence of C1qNAb, a dense plexus of h-Astrocytes is present near the injury epicenter. (**B**) A single epicenter injection of C1qNAb in acute SCI microenvironment dramatically decreases the proportion of h-astrocyte at the SCI epicenter. (**C**) Stereological analysis of STEM123+ h-Astrocytes in the spared tissue adjacent to the SCI epicenter, as well as in 1 mm regions rostral and caudal at 12WPT. Data show mean ± SEM (N = 6 mice per group). Statistical analysis using Student's t-test for comparison between vehicle + hNSC treated mice and C1qNAb + hNSC treated mice as indicated (*p<0.0261). (**D-F**) Catwalk kinematic gait analysis at 12 WPT shows that mice receiving C1qNAb + hNSC, but not vehicle + hNSC improves locomotor function in Catwalk kinematic testing vs. mice receiving only vehicle + vehicle. (**G**) Hindpaw duty cycle (% mean). (**H**) Catwalk Hindpaw stand mean (s). (**I**) Support three paws (%), indicating increased weight bearing. Data show mean ± SEM normalized to vehicle +vehicle-treated mice (as explained in *Supplementary file 1* only a subset of animals reached sufficient stepping parameters in this experiment to perform on CatWalk N = 3–5 mice per group as shown by individual points in graphs). Statistical analysis using Student's t-test for comparison between vehicle + hNSC treated mice and C1qNAb + hNSC treated mice as indicated.

The online version of this article includes the following figure supplement(s) for figure 8:

**Figure supplement 1.** C1q epicenter blockade does not alter lesion volume or hNSC engraftment after transplantation.

The cell surface C1q signaling/receptor proteins identified, CD44, GPR62, BAI1, c-MET, and ADCY5 are all of particular interest in the CNS. CD44 is a multi-ligand, multifunctional transmembrane glycoprotein involved in cell-cell and cell-matrix interactions, cell adhesion, migration and proliferation. CD44 is a receptor for hyaluronic acid (HA), but also interacts with other extracellular matrix ligands including osteopontin, laminin, and matrix metalloproteinases (*Ponta et al., 2003*; *Weber et al., 1996*). This protein participates in a wide variety of cellular functions mediated by alternative splice variations between CD44 isoforms, which confer ligand specificity (*Weber et al.,*

*1996*). CD44 has been linked to intracellular MAPK/ERK and PI3K/Akt signaling (*Herishanu et al., 2011*; *Zhang et al., 2010*), but it can also control cellular signaling through interactions with other cell surface receptors, e.g., receptor tyrosine kinases (RTKs), e.g., c-Met (also identified in our forward screen), G-protein coupled receptors like CXCR4, and Wnt-induced beta-catenin (*Schmitt et al., 2015*; *Orian-Rousseau and Schmitt, 2015*), which requires LRP6. High expression of CD44 has been linked to commitment as an astroglial progenitor (*Shaltouki et al., 2013*), and CD44 is highly upregulated in adult reactive astrocytes, suggesting a role in reactive astrogliosis (*Zamanian et al., 2012*). Oligoprogenitors/immature oligodendrocytes also express CD44 (*Zamanian et al., 2012*; *Naruse et al., 2013*). Transgenic mice overexpressing CD44 exhibit decreased oligodendrocyte maturation (*Liu et al., 2004*), and CD44 is required for oligoprogenitor migration after CNS transplantation (*Piao et al., 2013*). Critically, NSC also express CD44, as we show in fetal human brain under results, and CD44 mediates transendothelial migration of neural progenitor cells (*Deboux et al., 2013*; *Rampon et al., 2008*). Interestingly, CD44-HA interaction has been recently implicated in hippocampal neurogenesis, as a negative regulator of adult neural progenitor proliferation (*Su et al., 2017*).

Consistent with reported roles for CD44 on p-ERK signaling and cell migration in different cell types, in this study, we identified a novel ligand for CD44, C1q, and show that C1q-CD44 signaling specifically modulates ERK signaling and chemotaxis in hNSC in vitro. However, multiple of the C1q receptor candidates identified may signal in part via the ERK/MAPK pathway, and receptor activation is likely to be concentration-dependent, resulting in dramatic attenuation but not complete blockade of ERK signaling and migration for high C1q concentrations (C1q [200 nM]). Moreover, our data suggest that other signaling pathways (e.g., GPCR signaling for C1q modulation of hNSC proliferation) and candidate receptors may mediate different aspects of hNSC regulation by C1q. Concurrent activation of multiple signaling pathways, with specificity conferred by time, expression, and concentration, may also be biologically important.

In addition to CD44, we identified GPR62, BAI1, c-Met, and ADCY5 as hNSC cell surface interaction partners. GPR62 is an orphan, G-protein-coupled receptor (GPCR) that was identified based on a forward screen of novel GPCRs specific to the CNS and exhibits strong expression in essentially all areas of the brain (*Lee et al., 2001*). RNAseq data analysis on purified CNS cell types from adult brain indicate that GPR62 is highly expressed in mature oligodendrocytes (*Zhang et al., 2014*), consistent with our results from fetal brain single-cell RNA seq data analysis. GPR62 has not previously been investigated in the context of neural stem cell biology or immune interactions (*Khan and He, 2017*). Brain-specific angiogenesis inhibitor 1 (BAI1) is part of a small subfamily of adhesion GPCRs that includes its close homologs, BAI2 and BAI3, which are developmentally regulated and highly expressed in the CNS in both fetal and adult tissue. BAI1 and BAI3 were orphan GPCRs. However, phosphatidyl serine is now known to interact with BAI1 to mediate phagocytic clearance of apoptotic cells (*Barth et al., 2017*; *Park et al., 2007*). Additionally, BAI1 KO mice exhibit abnormal synaptic plasticity (*Zhu et al., 2015*) and deficits in hippocampal-dependent learning and memory, although normal synaptic arborization is maintained (*Zhu et al., 2015*). Interestingly, C1q-like molecules induce signal transduction through the TSR region of BAI3 (*Bolliger et al., 2011*). These data combined with the conserved homology of the TSRs across the BAI GPCR sub-family amplify interest in our preliminary identification of C1q as a BAI1 ligand. Critically, our data suggest that some C1q effects on hNSC are dependent on GPCR signaling, accordingly, future studies will investigate if GPR62 or BAI1 are involved in these observations. c-Met is an RTK, for which, hepatocyte growth factor (HGF) was the only known ligand. c-Met is expressed by cells of epithelial origin; however, in addition to the liver, c-Met expression is observed in cells of the cerebral cortex and choroid plexus (Allen brain atlas). Deregulation of c-Met expression in the adult is associated with malignancy, including cancers of the liver and brain, suggesting a role for c-Met in cell proliferation and expansion. HGF has been shown to drive NSC/progenitor cell migration via c-Met induced activation of PI3k/Akt in vitro (*Lan et al., 2008*; *Fang and Lee, 2014*). Further, HGF has been shown to act as both a proliferative factor and chemoattractant for subventricular zone NSC/progenitor cells (*Nicoleau et al., 2009*; *Wang et al., 2011*), an activity neutralized by both function blocking c-Met antibodies and expression of a dominant negative c-Met receptor (*Wang et al., 2011*). HGF-Met signaling in these studies also played a role in the maintenance of NSC/progenitors in an undifferentiated state; reduction in this signaling axis enhanced the generation of neurons by 7-fold. In parallel, HGF-c-MET signaling has been shown to increase the generation of oligoprogenitors/

oligodendrocytes from embryonic stem cells (*Hu et al., 2009*). ADCY5 belongs to the family of membrane-bound adenylate cyclases (AC), ADCY1-9. membrane-bound AC activity controls the cellular levels and subcellular localization of cAMP, and ADCY5/cAMP have been shown to regulate many physiological and pathophysiological processes through the modulation of downstream effectors, e.g., cAMP-dependent protein kinase A (PKA) (*Sassone-Corsi, 2012*), PI3K/Akt (*del Puerto et al., 2012*; *Okumura et al., 2007*), and ERK (*Yan et al., 2007*). Further studies will be necessary to determine the biological functionalities of these other novel ligand-receptor interactions.

In sum, this study identifies both novel functions for C1q as a single molecule in the CNS and a mechanism by which this protein directly modulates NSC through the identification of a novel C1q receptor, CD44. Moreover, in vivo genetic deletion of CD44 in hNSC restored the capacity of hNSC to drive locomotor recovery after acute SCI transplantation, expanding the therapeutic window to the acute injury microenvironment. Finally, these data broaden the potential biological role for inflammatory proteins and cells in the immune microenvironment to include a multitude of conditions in which complement and in particular C1q is upregulated in the context of the developing, aged, diseased, or injured CNS. Thus, understanding the molecular interplay and communication between neuroinflammatory molecules and NSC may be critical for predicting the effects of local cues on endogenous or donor cell properties. In all, these data support the potential for clinical application of strategies to target receptor candidates on transplanted cells rather than agents from the inflammatory microenvironment to enhance the repair or restoration of function by donor cells.

# Materials and methods

## Key resources table

| Reagent type (species) or resource | Designation | Source or reference | Identifiers | Additional information |
|---|---|---|---|---|
| Strain, strain background | Rag1 mice | JAXmice | 002216 - B6.129S7-Rag1 < tm1Mom>/J | |
| Strain, strain background | NODSCID mice | JAXmice | 001303 - NOD.Cg-Prkdc<scid > /J | |
| Cell line | Multipotent human CNS-derived stem cells | *Uchida et al., 2000* | Stem cells inc 2491.2 | Human Neural stem cells |
| Cell line | Human Neural stem cells | This paper and Piltti et al., in preparation | UCI161.1 | Human Neural stem cells (UCI) correspondence to aja@uci.edu |
| Transfected construct | Chimeric gRNA +hspCAS9 co-expression Vector | https://www.applied stemcell.com/ | Custom made | Guide RNA sequences in *Figure 5—figure supplement 1* |
| Antibody | Anti-STEM121 (mouse monoclonal) | TAKARA | Cat# Y40410 RRID:AB_2801314 | IF(1:3000) |
| Antibody | Anti-STEM123 (mouse monoclonal) | TAKARA | Cat# Y40420 Antibody ID:AB_2833249 | IF(1:3000) |
| Antibody | Fibronectin (rabbit) | Sigma Aldrich | Cat# F3648 RRID:AB_476976 | IF(1:500) |
| Antibody | Anti-C1q (mouse monoclonal) | Abcam | Cat#: ab71940 RRID:AB_10711046 | WB (1:200) Far-WB (1:50) PLA (1:100) |
| Antibody | Anti-pERK (rabbit) | Cell Signaling | Cat#: 4370 RRID:AB_10693765 | WB (1:1000) |
| Antibody | Anti-pAkt (rabbit) | Cell Signaling | Cat#: 9916 RRID:AB_10693765 | WB (1:1000) |
| Antibody | Anti-CD44 (rabbit monoclonal) | Abcam | Cat#: ab51037 RRID:AB_868936 | WB (1:1000) ICC (1:500) PLA (1:500) |
| Antibody | Anti-CD44 (rabbit) | Abcam | Cat#: ab189524 | ICC (1:500) |
| Antibody | Anti-c-MET (rabbit monoclonal) | Abcam | Cat#: ab51067 RRID:AB_880695 | PLA (1:500) |

*Continued on next page*

*Continued*

| Reagent type (species) or resource | Designation | Source or reference | Identifiers | Additional information |
|---|---|---|---|---|
| Antibody | Anti-BAI-1 (rabbit polyclonal) | Abcam | Cat#: ab135907 | PLA (1:500) |
| Antibody | Anti-GPR62 (rabbit polyclonal) | Thermo Fisher | Cat#: PA5-33745 RRID:AB_2551116 | PLA (1:500) |
| Antibody | Anti-ADCY5 (rabbit polyclonal) | Abcam | Cat#: ab66037 RRID:AB_1140781 | PLA (1:500) |
| Antibody | Anti-Olig2 (rabbit polyclonal) | Abcam | Cat#: ab136253 RRID:AB_2810961 | ICC (1:1000) |
| Antibody | Tubulin βIII (mouse monoclonal) | Biolegend | Cat#: MMS-435P RRID:AB_2313773 | ICC (1:500) |
| Antibody | GFAP (rabbit polyclonal) | Dako | Cat#: Z 0334 RRID:AB_10013382 | ICC (1:1000) |
| Antibody | Anti CD44 -PE | Miltenyi | Cat#:130-110-293 RRID:AB_2658165 | FACS (1:11) |
| Antibody | Anti CD133-FITC | Miltenyi | Cat#:130-113-673 RRID:AB_2726215 | FACS (1:11) |
| Antibody | Anti-C1qNAb (Goat-anti human) | Quidel | Cat#:A031 | C1q neutralization in vivo [100 μg/mL] 2 μL injection |
| Sequence-based reagent | Primers different mRNAs | This paper | NA | *Figure 5—figure supplement 1* |
| Peptide, recombinant protein | Purified human C1q | My Biosource | Cat. #: MBS143105 | [0.1nM = 28 ng/mL] [1nM = 400 ng/mL] [200nM = 80μg/mL] |
| Chemical compound, drug | PD98059 (pERK MAPK inhibitor) | GIBCO Life Technologies | Cat. #: PHZ1164 | 10 μM dissolved in DMSO |
| Chemical compound, drug | SB203580 (p38 MAPK inhibitor) | GIBCO Life Technologies | Cat. #: PHZ1253 | 10 μM dissolved in DMSO |
| Chemical compound, drug | Pertussis Toxin (GPCR inhibitor) | GIBCO Life Technologies | Cat. #: PHZ1174 | 10 μg/mL |
| Chemical compound, drug | DMSO | Sigma-Aldrich | Cat. #: D26650 | 1:1000 |
| Chemical compound, drug | PathScan Intracellular signaling array kit | Cell Signaling | Cat. #7744 | Phosphoarray Cell signalling |
| Chemical compound, drug | Pierce Pull-Down biotinylated protein-protein interaction kit; | Thermo | Cat. #21115 | Protein pulldown |
| Chemical compound, drug | RNeasy Mini Kit | Qiagen | Cat No./ID: 74104 | RNA extraction |
| Chemical compound, drug | DNA-free DNase I (Ambion) | Thermo Fisher | Cat. # AM2224 | DNA elimination |
| Chemical compound, drug | High Capacity cDNA Reverse Transcription Kit | ABI Life Technologies | Cat. # 4368814 | Retrotranscription |
| Chemical compound, drug | Duolink in situ Red starter kit Mouse/Rabbit | Sigma | Cat. # DUO92101 | Proximity Ligation Assay (PLA) |
| Chemical compound, drug | QCM Chemotaxis Cell Migration Assay, 96-well | Millipore/ Chemicon | Cat. # ECM510 | Cell migration |
| Software, algorithm | SnapGene | SnapGene | RRID:SCR_015052 | Sequencing |
| Software, algorithm | StereoInvestigator | Microbrightfield | | Stereology analysis |
| Software, algorithm | ImageJ | ImageJ | RRID:SCR_003070 | WB data analysis |

*Continued on next page*

*Continued*

| Reagent type (species) or resource | Designation | Source or reference | Identifiers | Additional information |
|---|---|---|---|---|
| Software, algorithm | Prism8 | Graphpad | RRID:SCR_002798 | Statistics and graphs |
| Other | Hoechst 33342 | Invitrogen Thermo | H3569 | 1:1000 |

## Animal models

All procedures involving animals were conducted in accordance with the Institutional Animal Care and Use Committee (IACUC) guidelines at University of California, Irvine (UCI). All animals used in SCI studies were females of 12–16 weeks of age and were on immunodeficient background, Rag1 (*Figure 7*, *Figure 7—figure supplements 1* and *2*) or NOD-scid (*Figure 8*, *Figure 8—figure supplement 1*) mice. Animal care, behavior, and analysis were performed by investigators blinded to groups, and random group allotment was used. Restricted randomization was applied to maintain equal group numbers. Pre-hoc exclusion criteria for stereological or behavioral assessments included: animals with abnormal scores (2 SD outside the cohort mean), unilateral bruising or abnormal force/displacement curves after contusion injury, or documentation of poor transplantation injection. Additionally, animals that exhibited a progressive decline in activity/function (N = 2) were excluded from the study, euthanized and autopsied. No evidence of infection or tumor formation was identified. Animal groups numbers were estimated in base of historical post-hoc power analysis of transplanted hNSC with stereological analyses (*Hooshmand et al., 2009*). Final group numbers used in histological or behavioral analysis are listed in *Supplementary file 1*.

## NSC lines and culture

All procedures using human cells were conducted in accordance with the Human Stem Cell Research Oversight (hSCRO) committee at UCI, and mNSC generation was approved by IACUC at UCI.

Multipotent human CNS-derived stem cells (referred to as hNSC) were isolated from 16 to 20 week gestation human fetal brain, and enriched for CD133 stem cell marker, which enhances the NSC population that exhibit neurosphere-initiating capacity by 2000-fold (*Uchida et al., 2000*). Cells derived in this manner retain multipotent capacity for over 20 passages in vitro and differentiate in a site-specific manner upon transplantation, generating neurons, oligodendrocytes, and astrocytes (*Uchida et al., 2000*). Two different cell lines were used in this study, with comparable derivation methods, Stem cells inc 2491.2 (*Figures 1–3* and *7*) and UCI161 (*Figures 4–6*). No substantial differences between the two different hNSC lines used in this study were observed in vitro, in terms of migration, CD133+ cell proportion, C1q novel receptor expression, and multipotency (Piltti et al., in preparation).

Mouse NSC (mNSC) were derived from single embryos cortices at E11.5–12. Cells were resuspended in DMEM media supplemented with the following ingredients (final concentration in parenthesis): N2 supplement (1×), B27 supplement (1×), bFGF (10 ng/mL), EGF (20 ng/mL), heparin (2 µg/mL), NAC (82 µg/mL), L-Glutamine (292 µg/mL), Na$^+$ Pyruvate (110 µg/mL) and cultured as small neurospheres for the first three passages, and then switched to hNSC media. hNSC/mNSC were cultured either as neurospheres or in monolayer in growth medium (GM): X-vivo 15 (Fisher) based media or Stemline (Sigma) based media supplemented with the following ingredients: N2 supplement (1×), bFGF (20 ng/mL), EGF (2 ng/mL), LIF (10 ng/mL), heparin (2 µg/mL) and NAC (63 µg/mL). Cultures were fed twice a week and passaged every 1–2 weeks.

For differentiation, cells were plated on poly-L-ornithine (PLO) (Sigma- 5 µg/mL) and recombinant mouse laminin (Invitrogen-10 µg/mL) coated 8-well glass chamber slides in X-vivo or Stemline differentiation medium (DM): supplemented with GDNF (10 ng/mL), BDNF (PeproTech) (10 ng/mL), bFGF (0.1 ng/mL), Ciprofloxacin (Cellgro) (10 µg/mL), Heparin (2 µg/mL), NAC (63 µg/mL), N2 (1:100) and B27 (1:20) for 14 d in vitro (hNSC), or 3–7 d (mNSC) as described previously (*Piltti et al., 2018*).

## C1q signaling

To assess C1q induction of intracellular signaling, hNSC were grown on 9.5 cm$^2$ multi-well plates at 90% confluence in GM. At 1 d post-plating cells were switched to DM with or without (control)

purified C1q at [0.1 nM], [1.0 nM], or [200 nM] concentrations for 15, 30, and 60 min as specified in figure legends. The cells were then collected and homogenized in cell lysis buffer (Cell Signaling Technology) and protein concentration was determined in precleared extracts using the micro BCA assay (Pierce). Phosphoarray analysis was performed with PathScan Intracellular signaling array kit, fluorescence readout (Catalog #7744, Cell Signaling technology) following manufacturers protocol, detecting simultaneously 18 phosphorylated molecules. Briefly, 150 µL of [200 µg/mL] protein lysates were added to the nitrocellulose-coated glass slides and incubated overnight at 4°C with gentle rocking motion, followed by exposure to the detection antibody cocktail for 1 hr room temperature (RT). Next, DyLight 680-linked streptavidin secondary antibody was added for 30 min at RT. The slides where analyzed in a LICOR CLx slide scanner and the fluorescence intensity measured for single spots. Samples were normalized to internal positive control spots and then to control conditions (DM without C1q). All experiments were performed using biological triplicates with technical duplicates.

To verify C1q induction of p-ERK and p-Akt signaling, cells were incubated with C1q for 60 min, total proteins were collected and homogenized with RIPA buffer (Sigma) containing protease (cOmplete ULTRA- Roche) and phosphatase (PhosSTOP-Roche) inhibitors. Equal amount of proteins (10 ug) were resolved in SDS-PAGE precast 4–12% gradient gels (NuPAGE Bis-Tris Gel; Life Technologies), transferred onto Nitrocellulose Pre-Cut Blotting Membranes NuPAGE (Life Technologies) in a semi-dry chamber (Trans-Blot SD Semi-Dry Cell, Bio-Rad), and immunoblotted. Antibodies against p-Akt, p-ERK, and β-actin were used for immunodetection. Bound antibodies were visualized using HRP-coupled secondary antibodies, followed by detection of labeled proteins using SuperSignal West Dura ECL kit (Pierce) and ChemiDoc imaging system (Bio-Rad). Optical band intensities were obtained using ImageJ software (National Institutes of Health; https://imagej.nih.gov/nih-image/). Band intensities were normalized to those of β-actin and subsequent normalization to untreated control conditions standardized at one and compared to control using one-sample t-test. All antibodies, sources, and dilutions are provided in *Supplementary file 2*. To assess p-ERK signaling pathway blockade, cells were grown as before and treated with p-ERK/MAPK pathway inhibitor (PD98059 −10 µM, GIBCO, catalog # PHZ1164) and with or without (control) C1q[200 nM] for 60 min. Samples were processed as explained above. All experiments were performed using biological triplicates.

## Far western blot analysis

To qualitatively determine the ability of C1q to interact with proteins from hNSC, a far-western blot was performed. In this assay, hNSC proteins from the total or cell surface fraction (below) were resolved in SDS-PAGE precast 4–12% gradient gels and transferred onto nitrocellulose pre-cut membranes in a semiwet chamber as described above. Immobilized proteins were incubated with or without C1q 25 µg/mL for 2 hr RT. Bound C1q was identified by imumunodetection, with anti-C1q antibody (monoclonal mouse anti C1q, 1:50, Abcam ab71940) 1 hr at room temperature, followed by anti mouse HRP-conjugated secondary antibody (ECL sheep anti-mouse IgG, HRP-linked whole antibody, 1:5,000, NA931V, GE Healthcare Life Sciences). Bound antibodies were visualized as described above.

Total protein was extracted with RIPPA buffer as described above. Cell surface proteins were purified using cell surface protein biotinylation assay. hNSC grown in monolayer were incubated with EZ-LinkSulfo-NHS-SS-Biotin (Thermo) following manufacturer's instructions. Briefly, cells were washed with PBS (supplemented with 0.1 mM $Ca^{+2}$ and 1 mM $Mg^{+2}$) 3 times for 5 min at 4°C and then incubated with biotin 1 mg/mL in PBS (0.1 mM $Ca^{+2}$ and 1 mM $Mg^{+2}$) for 30 min at 4°C. Biotin was then quenched with glycine 100 mM in PBS (0.1 mM $Ca^{+2}$ and 1 mM $Mg^{+2}$) for 30 min at 4°C. Cells then where washed with PBS and lysed with 200 µL/well Tris-HCl 50 mM, pH 7.5; NaCl 100 mM, Triton-100 0.5% v/v lysis buffer, with protease and phosphatase inhibitors cocktail (ROCHE), as described above for total protein extractions.

## C1q-protein interaction pull-down

To identify C1q binding proteins from hNSC total fraction, 100 µg of C1q was reversibly biotinylated with Ez-link Sulfo-NHS-SS-Biotinylation kit (Thermo) following the manufacturer's instructions. Briefly, to form C1q-biotin stock, 1 mg C1q (My Biosource) was resuspended in 2 mL Milli-Q $H_2O$ and buffer exchanged to PBS using a Zeba spin desalting column 7K MWCO (Thermo) following manufacturer's

instruction. Briefly, 200 µL of C1q stock (500 µg/mL = 0.25 nmol C1q) was incubated with a 20-fold molar excess of biotin reagent. 0.25 × 20 = 5 nmol of Sulfo-NHS-SS-Biotin for 2 hr 4°C in agitation. The un-bound biotin was eliminated by buffer exchange to PBS using a Zeba spin desalting column 7K MWCO (200 µL final volume). About 5 µL aliquot was used to determine C1q protein concentration by BCA analysis. A total of 60–80% of the original amount of C1q was recovered after the biotinylation process. An aliquot of the biotinylated C1q was analyzed by western blot to identify biotinylated C1q (*Figure 2—figure supplement 1A,B*); 50 µg of biotinylated C1q was utilized for pull-down experiments (Pierce Pull-Down biotinylated protein-protein interaction kit; Thermo). Briefly, 50 µg of biotinylated C1q (bait protein) was immobilized to a streptavidin agarose bead column for 30 min at 4°C with gentle agitation. The excess bait protein was removed by centrifugation, and then the not bound (available) streptavidin sites were blocked with free biotin (non-reactive). C1q-hNSC protein interactions were captured by incubation of immobilized C1q with 50 µg of hNSC total protein extract (prey proteins), for 2 hr at 4°C. Non-specific protein-interactions were removed by multiple washing steps. To finally elute and recover the attached proteins, the C1q attached to streptavidin columns was incubated with 1 mM DTT for 10 min, to break the SS bond of C1q to biotin (Sulfo-NHS-SS-Biotin) and recover all bound proteins. An aliquot of the elution was analyzed by SDS PAGE/silver staining and western blot to identify the bound proteins and recovered C1q (*Figure 2B*). Three experimental conditions were included in the pull-down experiments: Condition I: control, bait protein only, to verify C1q binding to the streptavidin column. Condition II: control, no bait protein, to identify unspecific interaction to the streptavidin column. Condition III: experiment, bait (C1q) and prey proteins (hNSC total protein lysate) to identify C1q hNSC-protein interaction. Bound proteins were eluted, trypsinized, and peptides identified using Mass Spectrometry (nanoLC-MS/MS) using a LTQ Orbitrap Velos Pro spectrometer [in the UCI Center for Virus Research Protein Mass Spectrometry Facility]. Proteins identified with high confidence in condition III, but not in conditions I and II, were considered to be associated with C1q specifically. Other controls were performed to validate this experimental procedure. Correct biotinylation of C1q was assessed by western blot analysis and ABC reaction (that identified only biotinylated C1q, *Figure 2—figure supplement 1A,B*). Binding ability of biotinylated-C1q versus naïve-C1q was assessed by far-western blot (*Figure 2—figure supplement 1C–E*).

## C1q cross-linking in live cells and pull-down

To assess C1q cell surface interactions in a biologically relevant setting where the 3D structure of ligand and receptor/s is maintained, we designed a different protein-protein interaction approach. In this strategy, cell surface interactions were assessed in live cells. Briefly, cells were grown in monolayer at 90% confluency, washed with (PBS 0.1 mM Ca+2 1 mM Mg+2) two times and incubated with 10 nM C1q in (PBS 0.1 mM Ca+2 1 mM Mg+2) at 4°C for 1 hr with gentle agitation. Bound C1q was cross-linked with a 20-fold molar excess of a cell impermeable cross-linker (Sulfo EGS Thermo) with an spacer arm of 16.1 A° to cross-link only closely interacting proteins following manufacturer's instructions. Sulfo EGS was added at a final concentration of 2 mM with bound C1q for 2 hr at 4°C. The cross-linker was quenched with quenching solution to a final concentration of 20 mM for 15 min at 4°C. Cells were then washed twice with (PBS 0.1 mM Ca+2 1 mM Mg+2 glicine 10 mM) and with PBS followed by cell lysis. Cells were lysed in Tris-HCl 50 mM, pH 7.5; NaCl 100 mM, Triton-100 0.5% v/v lysis buffer, with protease and phosphatase inhibitors cocktail (ROCHE) as described above. Pre-cleared extracts were then pulled down with immobilized anti C1q antibody ([JL-1] catalog # ab71940 Abcam). For the pull-down assay, C1q antibody was previously biotinylated with Ez-link Sulfo-NHS-SS-Biotinylation kit (Thermo) following the manufacturer's instructions and inmobilized as the bait protein in the Pierce Pull-Down biotinylated protein: protein interaction kit (Thermo) as described previously for C1q. To summarize, membrane proteins cross-linked with C1q were extracted and then pulled down with anti-C1q antibody. Conducted bound proteins were eluted and analyzed by mass spectrometry (nanoLC-MS/MS) at the UCI Center for Virus Research Protein Mass Spectrometry Facility, as described above.

## RNA extraction, and RT-PCR, PCR

Total RNA was isolated from cell culture samples using RNeasy Mini Kit (Qiagen), followed by elimination of genomic DNA using DNA-free DNase I (Ambion). For RT-PCR cDNA synthesis was

performed using High Capacity cDNA Reverse Transcription Kit (ABI Life Technologies) according to the manufacturers protocol. For each PCR reaction 200 ng of cDNA was amplified using GoTaq green Master mix (Promega), in thermal cycler conditions of 2 min 95℃, followed by 35 repeated cycles of 30 min at 95℃ 30 min at 58℃ and 1 min at 72℃ and a final elongation step of 10 min at 72℃. Primers used for C1q novel candidate receptor gene expression are listed in *Supplementary file 4*.

## Human embryonic cortex single-cell RNA-seq dataset analysis

Human embryonic cortex (weeks 22–23) single-cell RNA-seq dataset (GSE103723) was downloaded from GEO and processed according to the original publication (*Fan et al., 2018*) using Seurat (*Stuart et al., 2019*). Briefly, UMAP clustering was performed and all cells were classified into three major groups, namely, neurons, glial cells and non-neural based on global gene expression. Subsequently, glia cells were further processed and categorized into major glia types (e.g. Radial glia (NSC), Astrocyte Precursor Cells (APC)-Astrocytes, Oligoprogenitors (OPCs), and Oligodendrocytes) employing customized glia cell specific marker genes enrichment analysis implemented in hypeR (*Federico and Monti, 2019*). Glia cell type specific gene expression of the five candidate receptors were visualized by R package Pheatmap v1.012.

## Proximity ligation assay (PLA) for C1q interaction with signaling candidates

To verify C1q-candidate receptor protein interaction, hNSC were dissociated into single cells and plated on eight-well glass chamber slides in GM, 1 d post plating cells were incubated with purified C1q at 10 nM, 100 nM, or 200 nM concentration for 30 min as specified in figure legends. Cells were then fixed with 4% PFA for 15 min RT and cell membranes were stained with wheat germ agglutinin (WGA-Alexa conjugate 488- Invitrogen) (1:200 dilution) for 30 min followed by PLA analysis using the Duolink in situ Red starter kit Mouse/Rabbit (DUO92101, Sigma) according to the manufacturer's protocol. Briefly, cells were blocked for 30 min at 37℃ with blocking solution. Primary antibodies were incubated for 2 hr at room temperature (antibody concentrations and sources in *Supplementary file 2*), washed, then incubated with species-specific secondary antibody-complementary PLA probes (1:5) for 60 min at 37℃ in humidified chambers. PLA probes were then ligated for 30 min at 37℃, and amplified at 37℃ for 100 min. The slides were cover-sliped with Duolink in situ mounting media with DAPI (Sigma). For analysis, 13 random pictures per condition were captured using an ApoTome microscope system (Zeiss) as a total of 21 z-stacks of optical slices in 0.275 μm intervals using 40× objective. The number of red positive punctae was assessed using the 'analyze particles' function in the Image J analysis software, and reported relative to the total number of cells per picture frame. This quantification gives a rough number of average fluorescent particles per cell. All PLA data are shown as the average fluorescent punctae per picture relative to total number of cells per image (13 random images). All experiments were done with biological duplicates and technical duplicates.

## C1q induction of hNSC motility and chemotaxis

C1q induction of cell motility was evaluated in a live-cellimaging setting, mNSC were plated into custom-made PDMS microwells at density of 100 cells/microwell and imaged using VivaView FL Incubator Microscope (Olympus America, Inc) at +37℃ and 5% $CO_2$ with 20× objective in 20 min intervals for 7 d as previously described (*Piltti et al., 2018*). Time-lapse images of cells grown in DM, in the presence or absence (control) of C1q at [0.1 nM], [1.0 nM], and [200 nM] concentrations were compiled into movies and for each experiment 20 randomly selected individual cells (total of 60 cells per group) were manually tracked frame-by-frame using Imaris software 7.5.2 (Bitplane) as previously described (*Piltti et al., 2018*). Kinetics analysis was done using Imaris software to track cell motility over time. All data were normalized to control conditions. All experiments were conducted in biological triplicates.

C1q induction of hNSC chemotaxis was evaluated in transwell assays, hNSC and mNSC were grown as monolayer and dissociated as single cells. Source and concentration of C1q, and inhibitors used are indicated in *Supplementary file 3*. To perform the chemotaxis assay, single cells were resuspended in DM to a concentration of 300,000 cells/mL and then 100 μL of this single-cell

resuspension was added to each migration assay chamber, before placing chambers into feeder trays (Millipore) containing 150 µL of media for each condition; the chambers were then incubated at 37°C for 3.5 hr. Subsequently, the chambers were transferred onto new 96-well trays containing 150 µL of prewarmed cell detachment buffer and incubated for 30 min at 37°C. At the end of this incubation, 50 µL 1:75 dilution of CyQuant GR Dye:Lysis buffer was added to the cell detachment buffer and incubated for 15 min at room temperature. Finally, 150 µL CyQuant GR Dye:Lysis/detachment solution was transferred to a new 96-well plate, and migration was quantified using a 480/520 nm filter set on a fluorescent plate reader. To obtain fluorescence standardization, cells of known doses, as well as blanks containing only cell detachment buffer, lysis buffer, or CyQuant Dye, were used. All experiments were conducted in biological triplicate or quadruplicate with technical triplicates.

## Cell fate and proliferation analysis

To assess the effect of purified C1q on hNSC proliferation and differentiation hNSC were dissociated and plated as single cells on eight-well glass chamberslides. At 1 d post-plating, GM was changed to DM with or without (control) purified C1q at [0.1 nM], [1.0 nM], or [200 nM] concentrations with or without Brdu/EdU [10 mM] (proliferation). Cells were maintained in DM for 2 d in vitro (DIV) for proliferation and 14DIV for differentiation. hNSC were then fixed with 4% paraformaldehyde (PFA), permeabilized, and blocked in PBS solution supplemented with 0.1% Triton X-100 (Sigma-Aldrich), 5% goat or donkey serum (Jackson ImmunoResearch), and 1% BSA (Sigma-Aldrich). Primary and secondary antibodies including bromodeoxyuridine (Brdu), glial fibrillary acidic protein (GFAP), Olig2, and βIII-tubulin, sources, and the dilutions at which they were used are listed in *Supplementary file 2*. Hoechst 33342 (1:1000 dilution; Invitrogen) was used as a nuclear counterstain. Fluorescent images of immunostained slides were captured using random sampling with either an inverted Olympus IX71 fluorescent microscope or a ZEISS Axio Imager two light microscope with an Apotome2 image processor with 20× objective. Cell quantification was performed using Imaris software. All image capture and cell quantification was performed by investigators blinded to the study groups. All experiments in which hNSC were used were conducted in biological triplicate or quadruplicate with technical duplicates.

## CRISPR Cas9 hNSC CD44 genetic deletion

Our previous experimental data on hNSC behavior in vitro and in vivo after transplantation have utilized tissue-derived multipotent cells (*Cummings et al., 2005*; *Salazar et al., 2010*; *Sontag et al., 2014*; *Hooshmand et al., 2017*; *Nguyen et al., 2017*; *Cummings et al., 2006*; *Hooshmand et al., 2009*; *Piltti et al., 2013*), and not neuralized induced pluripotent or embryonic stem cells as a starting point. These cells are heterogenous at the time of gene editing, and isogenic clone selection carries a greatly heightened risk of inadvertent deletion of cell subpopulations or types. Accordingly, we developed a different strategy to derive CD44 WT and KO hNSC, using fluorescence activated cell sorting (FACS) to select WT and KO cell populations after CRISPR.

CD44 WT and KO hNSC were generated by gene editing with CRISPR Cas9 technology. Cas9 plasmid and guide RNAs were custom generated and validated in human HEK293 cells by applied stem cells https://www.appliedstemcell.com/. Briefly, two guide RNAs (gRNA) per gene were individually cloned into a bicistronic vector for co-expression with the Cas9 protein. Following vector delivery into HEK293 cells, the abilities of these gRNAS to guide Cas9 to target sites and promote double strand breaks were evaluated via Sanger sequencing (guide RNA sequences in *Figure 5—figure supplement 1A*). Briefly, hNSC were nucleofected at 80% efficiency (Lonza nucleofector buffer L) and co-transfected with two Cas9-GFP plasmids containing the gRNAs CD44g1 and CD44g5 to generate a 47 bp deletion of CD44 gene in exon 2. Unmodified hNSC exhibit 99.1% of CD44+ cells and 1.87% CD44- cells. Two weeks after CRISPR Cas nine modification, 39% of the cells remained positive for CD44 (WT) and 30% exhibited a negative expression (KO). CD44 WT and KO hNSC were then separated and enriched by fluorescence activated cell sorting (FACS). The gating parameters were highly stringent to avoid any heterozygous cell populations in our sorted cell pools (cells exhibiting low expression of CD44). CD44 expression was validated in WT and KO CRISPR Cas9-generated hNSC by flow cytometry (described below), total protein extraction and western blot, and Immunocytochemistry as described above, 2 weeks and 1–2 months after sorting respectively,

corroborating stable CD44 WT and KO expression. Antibodies, catalog numbers and dilutions used in *Supplementary file 2*. Two different CD44 KO lines were generated with the same method. No substantial differences between the two different cells lines generated regarding their CD133+ content, multipotency, proliferative capacity, differentiation profiles or responses to inflammatory components, like PMN-CM induced migration. Follow up experiments in vitro were done only for the transplanted cell line with biological triplicates or quadruplicates. Additionally, CD44 WT and KO cell lines were sequenced and genotype validating effective CRISPR Cas9 modification and homozygous generation of CD44 KO cell lines and maintenance of the WT genotype in the non-modified cells (*Figure 5—figure supplement 2*). Furthermore, CD133+ marker expression was unchanged in unmodified (WT) versus modified (KO) hNSC (*Figure 5—figure supplement 3C,D*), suggesting that CRISPR Cas9 modification did not substantially alter hNSC 'stemness'. No karyotype abnormalities (analyzed by cell line genetics https://www.clgenetics.com/) were observed in the WT and KO lines (*Figure 5—figure supplement 3A,B*).

## CD44 and CD133 flow cytometry

Cells grown in monolayer were dissociated as single cells and resuspended in PBS supplemented with 10% HSA (Octapharma); cells were then blocked with FcR blocking reagent (Miltenyi Biotec) and incubated with Pre-conjugated anti-CD44-PE (1:11, 130-110-293, Miltenyi Biotec) and anti-CD133/1 antibody (1:11, 130-113-673-FITC, Miltenyi Biotec) for 30 min at 4°C. After washes in PBS supplemented with 10% HSA the cells were analyzed using a BD LSR II Flow Cytometer (BD Biosciences). Cell viability was detected by propidium iodide (PI, 1:1000- Invitrogen) in unstained cells. In all conditions cell viability was typically more than 96%. Gating for all flow cytometry analyses were set using unstained controls, single labeled cells, and PI+ cells.

## Spinal cord contusion injuries for hNSC acute transplantations

Immunodeficient mice were anesthetized using 2.5% isofluorane and received a laminectomy at the thoracic vertebrae 9 (T9) using a surgical microscope. All animals received 50 kilodyne (kD; one dyne = 10 µN) contusion injury using the Infinite Horizon Impactor (Precision Systems and Instrumentation), as previously described (*Hooshmand et al., 2009*). For hNSC transplantation immunodeficient mice were used to enable long-term hNSC engraftment. NOD-scid, and Rag1 mice exhibit deficits in adaptive immunity, however, they demonstrate innate immune responses and histopathological characteristics comparable to other mouse strains following SCI (*Luchetti et al., 2010*). Number of animals per group/strain and exclusions are detailed in *Supplementary file 1*. hNSC were dissociated into a single-cell suspension and concentrated to a final density of 75,000 cells/µL in X-vivo media-(vehicle) on the day of SCI, and transplants were conducted, as described previously (*Hooshmand et al., 2009*; *Hooshmand et al., 2017*). Briefly, siliconized beveled glass pipettes (bevel: inner diameter = 70 µm, outer diameter = 100–110 µm; Sutter Instruments) were loaded with freshly triturated hNSC or vehicle, and injections were made into the intact parenchyma (two sites bilaterally above and below the injury epicenter) at T9 using a NanoInjector system and micropositioner (WPI Instruments) immediately after SCI contusion. Each site received 250 nL of cells or vehicle as described (*Cummings et al., 2005*). Previous studies in our laboratory have demonstrated that this volume does not exacerbate damage to the spinal cord.

## C1q neutralization in vivo

Immediately after injury, about 2 µL of C1q neutralizing antibody (C1qNab) goat anti-C1q [100 µg/mL], control goat anti-IgG antibody [100 µg/mL] (*Supplementary file 3*), or vehicle (media in which antibodies were prepared) were delivered via epicenter injections using a Hamilton Flexifil Syringe over 2 min, followed by a 1 min delay prior to withdrawal of syringe. The effect of C1qNAb at the concentration delivered in vivo was first tested in vitro, and no toxic or proliferative effects associated with exposure of hNSC to the C1qNAb was observed. Animals were randomly assigned to receive either C1qNAb with hNSC or vehicle with hNSC. The number of animals per group and exclusions are detailed in *Supplementary file 1*.

## Histology and stereological quantification

At 6WPI, 12WPI or 16WPI (*Supplementary file 1* for different animal cohorts) mice were terminally anesthetized, transcardially perfused with PBS, followed by 4% PFA. Spinal cord regions corresponding to dorsal roots at T2–T6, T6–T12, and T12-L2 were post-fixed and cryoprotected overnight in 4% PFA plus 20% sucrose and, flash frozen at −55°C in isopentane, and stored at −80°C until tissue processing (*Hooshmand et al., 2009*; *Hooshmand et al., 2017*). T6–T12 spinal sections were embedded in Neg50 Frozen Section Medium (Thermo) and 30 µm thick transverse-coronal sections where sectioned in a Cryostat with a CryoJane tape transfer system (Leica), as previously described (*Anderson et al., 1996*; *Sontag et al., 2014*). Frozen sections were collected on slides and stored at −20°C until processed for immunohistochemistry. All sectioning and immunohistochemistry procedures were conducted, as previously described (*Anderson et al., 1996*; *Sontag et al., 2014*). and dilutions used in immunohistochemistry are listed in *Supplementary file 2*. Unbiased histological quantification was conducted using stereology by investigators blinded to experimental groups. A random set of animals (n = 6/group) were selected for cell counts. The optical fractionator probe (*Joelving et al., 2006*) was used for estimation of the number of hNSC (STEM121+) and human astrocytes (STEM123+) in StereoInvestigator (Microbrightfield). Grid size measurements were determined based on preliminary experimentation to establish a low coefficient of error (CE ≤0.08). Cell migration was reported as the total estimated number of cells per 1 mm section, in a distribution graph was the injury epicenter is defined as the area of the cord with the largest Fibronectin positive area. A Cavalieri probe (grid size, 20 × 20 µm$^2$, MicroBrightField) was used to estimate total volumes of, spinal cord, fibronectin+ lesion volume, STEM121+, and STEM123+ cell clustering in 1 of 12 intervals from coronal spinal cord sections 360 µm apart, at 10× magnification.

## Detection of C1q at the SCI epicenter

A 4-month-old Rag1 female mice received either a laminectomy (control; n = 4) or a moderate contusion injury (50 kD; n = 12) at the thoracic level T9, as described below. Mice were PBS perfused 3 hr (n = 4), 24 hr (n = 4) or 9 d (n = 4) following SCI or laminectomy and the spinal cord freshly dissected and divided into 3 mm segments centered at T9, and two sequential 3 mm segments extending rostral and caudal from T9. Total protein extraction was performed with T-PER buffer (SIGMA) supplemented with protease (cOmplete ULTRA- Roche) and phosphatase (PhosSTOP-Roche) inhibitors. Protein concentration was determined in precleared extracts as described above. 10 µg of protein of the central (T9 or laminectomy) and the farthest rostral and caudal sections were resolved in SDS-PAGE precast 4–12% gradient gels, transferred onto nitrocellulose membranes and immunoblotted as described above. Antibodies against C1q and β-actin are listed in *Supplementary file 2*. Bound antibodies were visualized and analyzed as described above.

## Behavioral tasks and assessments of locomotor recovery

All behavioral data were collected and analyzed by blinded observers. Gross locomotor recovery in mice (n = 10/group; *Supplementary file 1*) was assessed using the Basso Mouse Scale (BMS; *Basso et al., 2006*) during the following time points: pre-injury and 2 d/7 d/2 weeks/4 weeks/6 weeks/8 weeks/12 weeks/16 weeks post-injury. Additionally, more sensitive behavioral parameters were utilized, including horizontal ladder beam and CatWalk gait analysis to assess kinematic parameters (*Koopmans et al., 2005*). Horizontal ladder beam and CatWalk tasks were performed preinjury and every 4 weeks after injury until animals were sacrificed 16 weeks post-injury.

## Statistics

For in vitro assays, experiments were performed with biological replicates and technical replicates, as stated in each method section or figure legend. Biological replicates are defined as experiments performed independently from different cell samples (e.g., different cell culture, passage). Technical replicates are defined as experiments performed in duplicates or triplicates with the same cell samples (e.g., same culture sample). We used N3-4 biological replicates consistent with accepted standard for in vitro studies. Statistics were performed using Prism, version 8.0 (Graphpad Software Inc, San Diego, CA, http://www.graphpad.com). All data are presented as mean ± standard error of the mean (SEM). When indicated experimental values were normalized to untreated control conditions and standardize to 1 or 100%. Data were analyzed using one-sample t-test to compare with control

and Student's t-test to compare between experimental conditions. When comparing across multiple groups, one-way ANOVA was conducted first, followed by Tukey's multiple t-test comparisons. In all statistical analyses, significance was defined as $p \leq 0.05$.

## Acknowledgements
The authors are grateful to Rebecca Nishi, MS, Javier Lepe, BS, Chris Nelson, BA, Hongli Liu, MD, for assistance with SCI surgeries and BMS behavioral open field test. The authors thank students Karishma Kumar, Jason Bahremand, BS, Rami Fadi Halaseh, BS, and Phoebe Bui BS, for cell quantification and behavioral analysis. This work was supported by NIH shared instrumentation grant to PDG, as well as grants from the Craig H Nielsen Foundation and the Christopher Reeve Foundation to AJA.

## Additional information

### Funding

| Funder | Grant reference number | Author |
| --- | --- | --- |
| Craig H. Neilsen Foundation | CHN-316291 | Aileen Anderson |
| Christopher and Dana Reeve Foundation | AAC-2013(5) | Aileen Anderson |
| National Institutes of Health | 1S10OD016328-01 | Paul D Gershon |

The funders had no role in study design, data collection and interpretation, or the decision to submit the work for publication.

### Author contributions
Francisca Benavente, Conceptualization, Data curation, Formal analysis, Supervision, Investigation, Methodology, Writing - original draft, Project administration, Writing - review and editing; Katja M Piltti, Mitra J Hooshmand, Data curation, Investigation, Methodology; Aileen A Nava, Investigation, Methodology; Anita Lakatos, Data curation, Software, Methodology; Brianna G Feld, Dana Creasman, Investigation; Paul D Gershon, Resources, Methodology; Aileen Anderson, Conceptualization, Resources, Supervision, Funding acquisition, Methodology, Writing - original draft, Project administration, Writing - review and editing

### Author ORCIDs
Francisca Benavente (iD) https://orcid.org/0000-0001-8245-2673
Mitra J Hooshmand (iD) http://orcid.org/0000-0001-6159-6934
Aileen A Nava (iD) http://orcid.org/0000-0002-2053-6713
Aileen Anderson (iD) https://orcid.org/0000-0002-8203-8891

### Ethics
Animal experimentation: All procedures involving animals were conducted in accordance with the Institutional Animal Care and Use Committee (IACUC) guidelines at University of California, Irvine (UCI). protocols AUP-17-071, AUP 17-115 All procedures using human cells were conducted in accordance with the Human Stem Cell Research Oversight (hSCRO) committee at UCI, hSCRO: protocols 2006-5294 for In vivo transplants, 2007-5493 for in vitro work. mNSC generation was approved by IACUC at UCI. Protocols AUP 17-115.

### Decision letter and Author response
Decision letter https://doi.org/10.7554/eLife.55732.sa1
Author response https://doi.org/10.7554/eLife.55732.sa2

## Additional files

### Supplementary files

- Supplementary file 1. Number of animals per group for in vivo studies.
- Supplementary file 2. List of antibodies and concentrations used for immunodetection.
- Supplementary file 3. List of conditions used in vitro for migration, proliferation, fate, and signaling assay.
- Supplementary file 4. List of primers used in this manuscript.
- Transparent reporting form

### Data availability

All data generated or analyzed during this study are included in the manuscript and supporting files.

The following previously published dataset was used:

| Author(s) | Year | Dataset title | Dataset URL | Database and Identifier |
|---|---|---|---|---|
| Fan X, Dong J, Wang X, Qiao J, Tang F | 2018 | Single cell RNA-seq analysis of human embryonic cortex | https://www.ncbi.nlm.nih.gov/geo/query/acc.cgi?acc=GPL20301 | NCBI Gene Expression Omnibus, GSE103723 |

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
