## [Decision Letter]

**Acceptance summary:**

The authors have discovered a new pathway for regulation of neural stem cells, cells that can give rise to neurons, myelin producing cells and supporting cells of the brain. The findings may yield new approaches to brain repair using stem cell transplants.

**Decision letter after peer review:**

Thank you for submitting your article "Novel C1q receptor-mediated signaling controls neural stem cell behavior and neurorepair" for consideration by *eLife*. Your article has been reviewed by two peer reviewers, and the evaluation has been overseen by a Reviewing Editor and Marianne Bronner as the Senior Editor The following individuals involved in review of your submission have agreed to reveal their identity: Daniel Cortes (Reviewer #2); Mark Noble (Reviewer #3).

The reviewers have discussed the reviews with one another and the Reviewing Editor has drafted this decision to help you prepare a revised submission.

Summary:

The study provides insight into a novel signaling mechanism based on the complement protein C1q for the control of neural stem cell migration and differentiation, findings which are highly relevant to repair in the central nervous system.

Essential revisions:

Both reviewers are very enthusiastic about the findings in this study. Reviewer 1 points to some areas in two figures that require clarification or further explanation.

When you compare the results of Figure 7I vs. Figure 8C and specifically controls, they look really different. I understand inevitable variability among experiments, but why do 123 positive cells drop abruptly out of the epicenter on Figure 8C and not on Figure 7I? I also notice that those are different values. Why is one expressed in total numbers, and the other one is expressed in percentage? That is confusing, is there a reason for that? That makes them hard to compare, but in any case, I don't think that accounts for the observed variability between the two plots. I suggest that the experiment in Figure 8C must be done with another control. The final experiment should be CD44-WT, CD44-KO, and C1q Nab + CD44WT; unless they can correct this disparity. The full reviews are enclosed for clarity.

Reviewer #1:

The authors have previously found that C1q modulates NSC proliferation, fate, and migration. Here they want to discover the signaling pathway for such effects. They found that five different membrane receptors bind C1q, and specific downstream proteins are phosphorylated upon the addition of C1q. The experiments support that CD44 mediates migration and gliogenesis; nonetheless, the proliferation of NSCs is a CD44-independent process. They finally found that after spinal cord injury, CD44-KO improves spinal cord injury recovery. The authors uncovered a novel signaling mechanism by which C1q modulates NSCs activity. The previous effect seems to be independent of the immune system finding new roles for C1q.

1) If you compare the results of Figure 7I vs. Figure 8C and specifically controls, they look really different. I understand inevitable variability among experiments, but why do 123 positive cells drop abruptly out of the epicenter on Figure 8C and not on Figure 7I? I also notice that those are different values. Why is one expressed in total numbers, and the other one is expressed in percentage? That is confusing, is there a reason for that? That makes them hard to compare, but in any case, I don't think that accounts for the observed variability between the two plots. I suggest that the experiment in Figure 8C must be done with another control. The final experiment should be CD44-WT, CD44-KO, and C1q Nab + CD44WT; unless they can correct this disparity.

This is a really well-conducted project, with clear goals and novel results. Overall, the paper is solid enough to support the publication after the suggested changes.

Reviewer #2:

It is a delight to read such a well thought out and interesting manuscript. This was a well-written, indeed a gripping, read from start to finish. The findings are fascinating, the experiments are very well conducted, and the story is artfully told.

The core of this story is that the authors previously found a non-immune role for complement C1q in modulating human neural stem cell (hNSC) migration and fate. They now have investigated the mechanism of C1q's effects, and show that it has the surprising property of being a ligand that activates intracellular signaling pathways. They identified five transmembrane C1q signaling/receptor candidates in hNSCs (CD44, GPR62, BAI1, cMET, and ADCY5). A deeper investigation revealed that CD44 mediates C1q induced hNSC signaling and chemotaxis in vitro. They also show that this pathway mediates hNSC migration and functional repair in vivo after spinal cord injury and that manipulation of this pathway can enhance the therapeutic benefits of nHSC transplantation.

The idea that C1q has non-traditional roles has been shown in previous studies by this group and others, and it is clear that there are C1q effects that are separate from complement cascade activation. These additional properties suggest that C1q is a ligand for something, but it is not known how it works in this regard.

After demonstrating that C1q activates MAP/ERK and PI3K pathways at nM concentrations, they used an unbiased screen assay to cross-link C1q to surface proteins on hNSCs. This approach yielded five candidate receptors, which were validated using elegant Proximity Ligation Assays. They next showed that inhibition of pathway activation with pharmacological inhibitors blocked effects of C1q.

They focused on CD44 due to previous indications of the effects of CD44 on ERK signaling pathways (although the same would be true for c-MET). This was a good choice, as CD44 knockout cells did not show ERK activation or stimulation of migration in response to C1q. Similar results were obtained with NSCs derived from CD44 knockout mice.

They then conducted in vivo studies and showed that CD44 knockout hNSCs show differences in their localization in traumatic SCI, differences in astrocyte generation and enhancing the ability of hNSCs to promote locomotor recovery in acute SCI. These data indicate effects of C1q in vivo. They also found spatial and temporal C1q gradients in vivo following SCI, with a distribution consistent with results of their other studies. They also showed improved motor recovery, thus demonstrating a potential therapeutic utility of their findings. C1q depletion had similar effects as CD44 knockout. This was done by injecting a C1q neutralizing antibody into the lesion center.

It is also exciting that this is likely to be the beginning of a story that will continue to become still more interesting.

As expected for this team, the data are all rock solid.

In short, this is wonderfully exciting paper. Congratulations to the authors on such a lovely piece of research.

---

## [Author Response]

Essential revisions:Both reviewers are very enthusiastic about the findings in this study. Reviewer 1 points to some areas in two figures that require clarification or further explanation.When you compare the results of Figure 7I vs. Figure 8C and specifically controls, they look really different. I understand inevitable variability among experiments, but why do 123 positive cells drop abruptly out of the epicenter on Figure 8C and not on Figure 7I? I also notice that those are different values. Why is one expressed in total numbers, and the other one is expressed in percentage? That is confusing, is there a reason for that? That makes them hard to compare, but in any case, I don't think that accounts for the observed variability between the two plots. I suggest that the experiment in Figure 8C must be done with another control. The final experiment should be CD44-WT, CD44-KO, and C1q Nab + CD44WT; unless they can correct this disparity. The full reviews are enclosed for clarity.

We thank the reviewer for highlighting the difference in data presentation for Figure 7I and Figure 8C. This has been addressed, and the data in Figure 7I and Figure 8C have been re-graphed to be in identical format (stereological estimate of SC123 cell number). We have modified both the text and the figure legends to include these changes in the paper.

We also note several additional points for the reviewer’s consideration. First, while the contusion injury induced was theoretically the same, these experiments were separated by a large gap in time. Second, while both experiments were conducted in constitutively immunodeficient mouse strains, these were different due to availability (Rag1 – Figure 7 and NODScid mice Figure 8). Finally, the timepoints of analysis were not identical (16 WPT Figure 7I and 12WPT Figure 8C). Accordingly, one would not expect an exact overlay between datasets – however, despite these differences, the fact that transplanted hNSC behaved similarly, including in total cell baseline engraftment (Author response image 1) does indicate that the hNSC transplants and cell behavior between these experiments was generally quite consistent.

**Author response image 1. sa2fig1:**